# *TP53* exon-6 truncating mutations produce separation of function isoforms with pro-tumorigenic functions

Nitin H Shirole[1,2], Debjani Pal[1,3], Edward R Kastenhuber[4], Serif Senturk[1], Joseph Boroda[1], Paola Pisterzi[1], Madison Miller[1], Gustavo Munoz[1], Marko Anderluh[5], Marc Ladanyi[4], Scott W Lowe[4], Raffaella Sordella[1,2,3]*

[1]Cold Spring Harbor Laboratory, Cold Spring Harbor, United States; [2]Graduate Program in Genetics, Stony Brook University, Stony Brook, United States; [3]Graduate Program in Molecular and Cellular Biology, Stony Brook University, Stony Brook, United States; [4]Department of Cancer Biology and Genetics, Memorial Sloan Kettering Cancer Center, New York, United States; [5]Department of Medicinal Chemistry, University of Ljubljana, Ljubljana, Slovenia

**Abstract** *TP53* truncating mutations are common in human tumors and are thought to give rise to p53-null alleles. Here, we show that *TP53* exon-6 truncating mutations occur at higher than expected frequencies and produce proteins that lack canonical p53 tumor suppressor activities but promote cancer cell proliferation, survival, and metastasis. Functionally and molecularly, these p53 mutants resemble the naturally occurring alternative p53 splice variant, p53-psi. Accordingly, these mutants can localize to the mitochondria where they promote tumor phenotypes by binding and activating the mitochondria inner pore permeability regulator, Cyclophilin D (CypD). Together, our studies reveal that *TP53* exon-6 truncating mutations, contrary to current beliefs, act beyond p53 loss to promote tumorigenesis, and could inform the development of strategies to target cancers driven by these prevalent mutations.

*For correspondence: sordella@cshl.edu

**Competing interests:** The authors declare that no competing interests exist.

## Introduction

The International Cancer Genome Consortium has recognized that the *TP53* gene is the most frequently mutated gene in human cancer (*Hollstein et al., 1991*; *Olivier et al., 2010*). Genetic studies show that, in most tumors, *TP53* point mutations co-occur with the loss of one copy of the *TP53* gene (LOH) due to deletions in chromosome 17 where the *TP53* locus is located (*Baker et al., 1989*; *Menon et al., 1990*; *Olivier et al., 2010*; *Rivlin et al., 2011*; *Liu et al., 2016*). Consistent with these observations and the two-hit hypothesis proposed by A.G. Knudson, experimental evidences have led to the description of *TP53* as a tumor suppressor gene (*Knudson, 1971*; *Baker et al., 1989*; *Finlay et al., 1989*; *Donehower et al., 1992*).

This simplistic vision has been challenged by recent studies spurred by the observation that *TP53* missense mutations do not have a uniform distribution; rather, they occur more frequently at specific residues (R175, G245, R248, R249, R273 and R282) often referred to as 'hotspot' *TP53* mutation sites (*Petitjean et al., 2007*; *Brosh and Rotter, 2009*). The high frequency of these mutations led to the hypothesis that these hotspot mutations could not only result in loss of function activities, but also could confer an advantage of growth to cancer cells. Indeed, many lines of evidence have now demonstrated that certain p53 missense mutants could exhibit a gain of function activities during tumorigenesis (*Brosh and Rotter, 2009*; *Oren and Rotter, 2010*). For instance, some of the gain of function mutations, including R175H, R248Q, R273H, resulted in an increase in cell invasion, cell

migration, cell proliferation and anti-apoptosis in different in-vitro models (*Muller and Vousden, 2014*). Additionally, mice expressing *TP53* R172H (human R175H) and R270H (human R273H) mutations manifest a broad spectrum of aggressive tumors that are more metastatic in nature when compared to p53-null mice (*Lang et al., 2004*; *Olive et al., 2004*; *Doyle et al., 2010*). Though different gain of function mutants exhibit various pro-tumorigenic phenotypes, their mechanism of function mostly relies on alterations to the p53 transcription program (*Freed-Pastor and Prives, 2012*).

In this study, we similarly report that certain *TP53* truncating mutations promote tumorigenesis rather than halt it. In fact, we observed that *TP53* exon-6 truncating mutations occur at higher than expected frequencies and, when ectopically expressed in cells, induce the acquisition of pro-metastatic features. In contrast to *TP53* missense gain of function mutations, we found that *TP53* exon-6 truncating mutations are necessary for cell survival in normal 2-D cell growing conditions. These *TP53* truncating mutations also different from a canonical p53 missense gain of function mutants in regards to their mode of action. As we have shown in this study, these p53 mutants lack transcriptional activity and, instead, have phenotypes that depend on their molecular and functional interactions with Cyclophilin D in the mitochondria.

Much like EGFR, ROS and ALK mutations have been candidates for precision medicine, the relatively frequent distribution of exon-6 *TP53* truncating mutations in certain tumors, combined with the availability of CypD inhibitors, implies that these mutations may similarly be successfully targeted with precision medicine.

## Results

### *TP53* exon-6 truncating mutations occur at a higher than expected frequency

While the gain of function activity of p53 missense mutants has been studied extensively (*Brosh and Rotter, 2009*; *Oren and Rotter, 2010*), the biological effects of p53 nonsense mutants have yet to be explored.

To address this, we first examined a panel of 22 sequencing studies, predominantly carried out by the Cancer Genome Atlas (TCGA) project and accessed using the cBio portal, referred to here as the 'TCGA cohort' (*Cerami et al., 2012*). Studies were selected for inclusion on the basis of having more than 100 samples per tumor type and at least 10 tumors with *TP53* mutations (*Supplementary file 1*). As shown in *Figure 1A*, it is evident that *TP53* nonsense mutations are distributed non-randomly with increased frequency in correspondence to *TP53* exon-6 (i.e. *TP53* exon-6 nonsense). Interestingly, our analysis also indicated that nonsense mutations occur at sites distinct from those affected by missense mutations (*Figure 1A*). These findings were confirmed in an independent pan-cancer dataset of 3797 cases, in which targeted sequencing was performed at Memorial Sloan Kettering Cancer Center (*Cheng et al., 2015*), referred to here as the 'MSKCC cohort' (*Figure 1B*).

To determine the relative strength of selection pressure for nonsense and missense mutants, we calculated the number of theoretical changes (i.e., expected cases) and compared this value to the observed frequency of mutations. As indicated in *Figure 1E*, we found that in the case of missense mutations the relative expected frequency was 0.34, while the relative observed frequency was 0.64, based on the analysis of the TCGA cohort. This finding suggests that missense mutations are observed at a frequency that is 1.9 fold higher than expected. In the case of nonsense mutations, we observed a slight difference between the expected frequency (i.e., 0.054) and observed frequency (i.e., 0.084) of mutations. Interestingly, when we limited our analysis only to exon-6 nonsense mutations we found a 4–5 fold increase in the observed frequency compared to the expected frequency (*Figure 1C and D* and *Figure 1—figure supplement 1A*). This finding indicates that nonsense mutations in exon-6 occur nearly 5 times more frequently than all *TP53* mutations (p=3.869e-11) and 2–3 times more frequently than either missense mutations or nonsense mutations outside of exon-6 (p=3.71e-4 and p=8e-5, respectively). These results were confirmed in the MSKCC cohort (*Figure 1D and E* and *Figure 1—figure supplement 1B*).

In addition to nonsense mutations, frameshift and splice site mutations may produce truncated proteins. We found that even though nonsense mutations occur at a rate of 13%, overall truncating mutations account for more than 25% of all *TP53* mutations in both datasets examined (*Figure 1—*

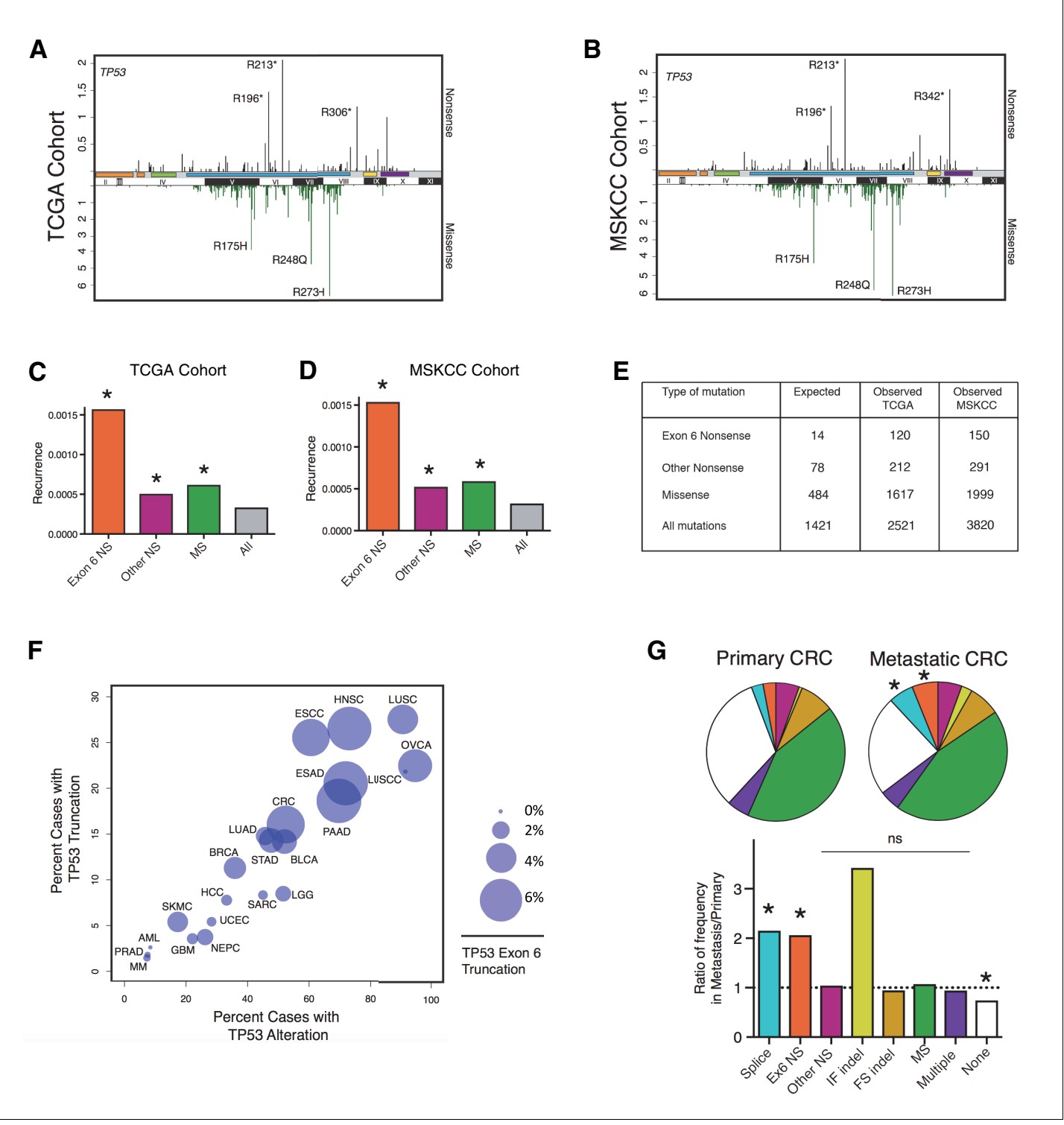

**Figure 1.** *TP53* exon-6 truncating mutations occur at higher than expected frequency. (A) Distribution of *TP53* nonsense (top, black) and missense (bottom, green) mutations in the TCGA cohort (n = 2521 tumors). Domains are demarcated on the upper baseline as follows: trans-activating domains (orange), Proline rich domain (green), DNA binding domain (light blue), nuclear localization sequence (yellow), and oligomerization domain (purple). The lower baseline and Roman numerals below indicate *TP53* exon location relative to the p53 coding sequence. (B) Analysis as in A in the MSKCC cohort (n = 3797 tumors). (C–D) Recurrence frequency of each mutation type per unique change per sample in the TCGA and MSKCC cohorts respectively (Missense, p<2.2e-16; other nonsense, p=0.00178 and exon-6 nonsense, p=3.869e-11, Fisher's exact test). (E) Count of unique reported amino acid changes and observed instances of exon-6 nonsense, other nonsense, missense, or all mutations. (F) Frequency of *TP53* alteration vs.
*Figure 1 continued on next page*

*Figure 1 continued*

frequency of *TP53* truncating mutations by cancer type. Circles were plotted in proportion to the frequency of *TP53* exon-6 truncation mutations. See *Supplementary file 1* for cancer type abbreviations. (G) The pie charts represent the relative frequency of *TP53* mutation type for colorectal cancer primary tumors (top left, n = 403) and metastases (top right, n = 395). Mutations are indicated as follows: splice site mutations (Splice, light blue, p=0.035, fisher's exact test), exon-6 nonsense mutation (Ex6 NS, orange, p=0.041), other nonsense mutation (Other NS, pink), in-frame insertion/deletion (IF indel, yellow), frameshift insertion/deletion (FS indel, gold), missense mutation (MS, green), multiple mutations (Multiple, purple), or no *TP53* mutation (None, white). Note that both exon-6 nonsense mutations (p-value = 0.041) and missense mutations (p-value = 0.035) are over-represented in the metastatic samples with respect to p53-WT cases, whereas nonsense mutations outside of exon-6 and missense mutations are not (Fisher's exact test). The lower chart indicates the ratio of frequency in metastases to primary colorectal cancers of the indicated *TP53* mutations. See *Supplementary file 3* for number of tumor samples with *TP53* mutation in primary CRC and metastatic CRC.

The following figure supplements are available for figure 1:

**Figure supplement 1.** Distribution of *TP53* somatic mutations across multiple tumor types based on TCGA and MSKCC data set analysis.

**Figure supplement 2.** Graphical summary of *TP53* non-synonymous mutations in melanoma in indicated studies.

**Figure supplement 3.** *TP53* exon-6 truncating mutations are distributed at different frequency in different tumors.

*figure supplement 1*). We also observed that the distribution of *TP53* mutations differs in various tumor types, ranging from 7.3 to 94.9% in the case of multiple myeloma (MM) and ovarian serous cystadenocarcinoma (OVCA) respectively (*Supplementary file 1*). As for *TP53* truncating mutations, their frequency of occurrence spans from 1.46% in MM to 27.53% in lung squamous cell carcinoma (LUSC). Notably, in pancreatic adenocarcinoma (PAAD), esophageal adenocarcinoma (ESAD), squamous cell carcinoma (ESCC), head and neck squamous cell carcinoma (HNSC), colorectal adenocarcinoma (COAD) and skin cutaneous melanoma (SKCM), the frequency of *TP53* exon-6 truncating mutations was higher than 6% (*Supplementary file 1*).

Although the frequency of *TP53* exon-6 truncating mutations followed the distribution of *TP53* mutations in the majority of tumors examined, this was not always the case. This phenomenon is best exemplified by our observations of lung small cell carcinoma (LUSCC) wherein, despite the nearly ubiquitous presence of *TP53* alterations, we found no exon-6 truncations (*Figure 1F* and *Figure 1—figure supplement 3*).

Strikingly, in the MSKCC cohort, we found a statistically significant increase in the frequency of *TP53* exon-6 truncating mutations in colorectal cancer (CRC) metastatic site tumors with respect to primary tumors (*Figure 1G* and *Supplementary file 3*). This is reminiscent of a previous study in which an analysis of colorectal cancers revealed an increased representation of *TP53* exon-6 mutations in liver metastases (*Miyaki et al., 2002*). Although we found an increase in In-Frame indel mutations in metastatic tumors compared to primary tumors, this increase was not deemed statistically significant as the number of primary tumors identified with mutations was very low (n = 3 for primary and n = 10 for metastasis).

## p53 exon-6 truncating mutants reprogram cells towards the acquisition of pro-metastatic features

In principle, a higher than expected occurrence of truncating mutations could be linked to a particular etiology, to variable nucleotide substitutions or to translation termination efficiency (*Mort et al., 2008*). On the other hand, and as has been demonstrated for a gain of function *TP53* mutations, higher than expected frequency of *TP53* exon-6 truncating mutations could instead underlie a selective advantage during tumorigenesis (*Muller and Vousden, 2014*).

To test this hypothesis, we generated cell lines ectopically expressing multiple p53 C-terminal truncated proteins mirroring the R213* and R196* exon-6 p53 truncating mutants, and compared their activities to p53 full length (i.e., p53-WT), a longer p53 truncating mutant (G325*) and a shorter p53 truncating mutant (W146*) (*Figure 2A and B* and *Figure 2—figure supplement 2*).

As it is consistent with their lack of tumor suppressor activities, different p53 C-terminal truncating proteins ectopically expressed in a p53 homozygous deletion cell line (H1299) failed to decrease cell viability (*Figure 2C*). Yet, as shown in *Figure 2*, ectopic expression of *TP53* exon-6 truncations in

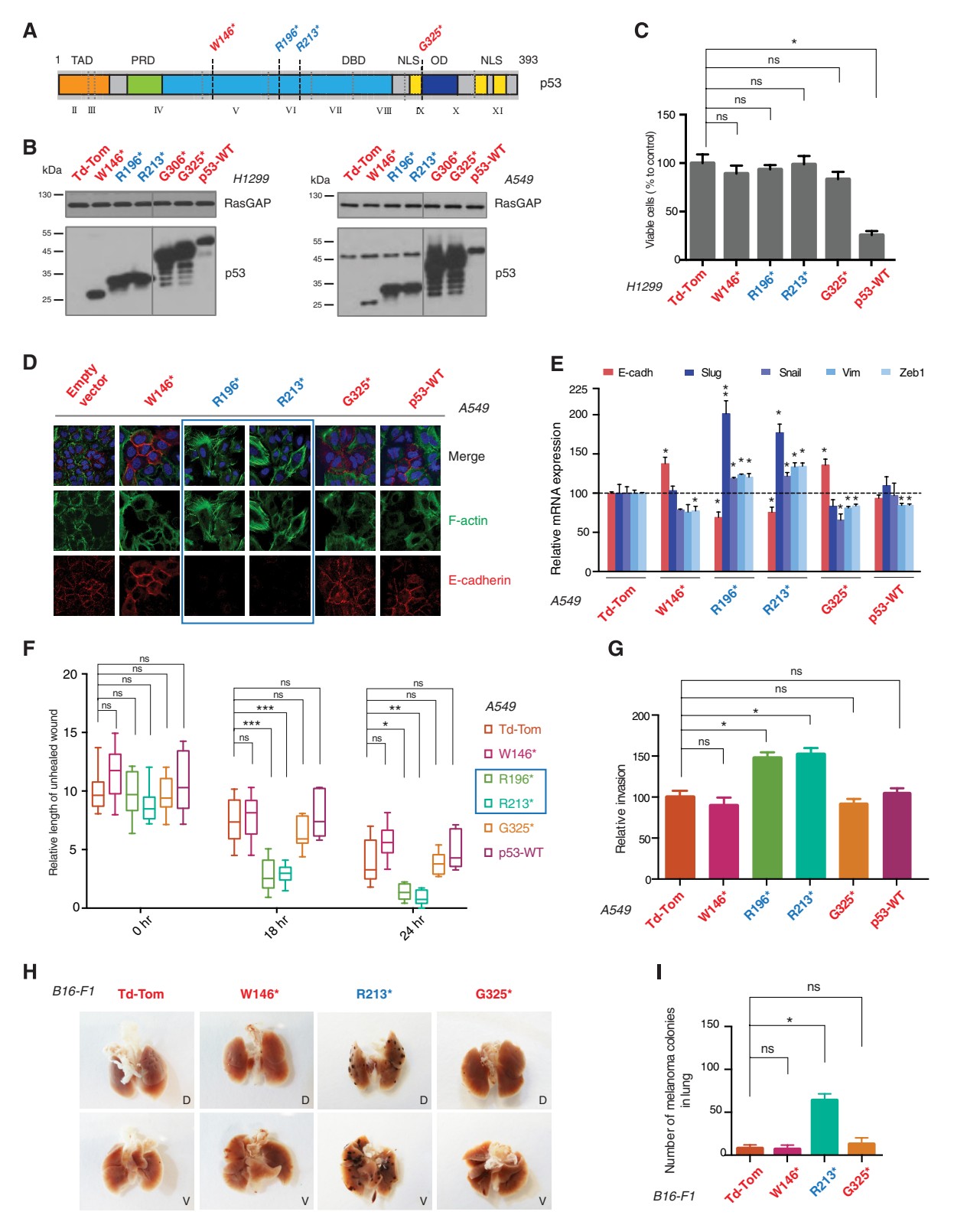

**Figure 2.** p53 exon-6 truncating mutants reprogram cells towards the acquisition of pro-metastatic features. (**A**) Schematic of p53 and position of p53 nonsense mutations utilized in this study. Domains are demarcated and *TP53* exon locations relative to the p53 coding sequence are indicated in the Roman numerals. (**B**) Different p53 truncations were ectopically expressed in the p53 null H1299 and p53-WT A549 cell lines. Expression was verified by western blot analysis of cell extracts by p53 N-terminal specific (DO1) antibody and the RasGAP as loading control after 48 hr post infection. (**C**) p53

*Figure 2 continued on next page*

*Figure 2 continued*

C-terminal truncations lack tumor suppressor capabilities. The chart indicates the number of viable cells relative to Td-Tom expressing cells at 72 hr post infection. Each bar is the mean of 9 replicates (p-value *<0.0005 unpaired t-test). (D) Immuno-staining of A549 cells with phalloidin (green), E-cadherin (red) and DAPI (blue). Note that cells expressing R196* and R213* are characterized by different morphology, presence of stress fibers and decreased expression and localization of E-cadherin. (E) RT-qPCR analysis of EMT markers in A549 cells expressing different p53 truncations. mRNA expression was quantified by SYBR-green-based RT-qPCR. Each bar is the average of 3 replicates and represents the mRNA expression of the indicated genes relative to GAPDH (p-value *<0.05 and **<0.005, unpaired t-test). (F) p53 exon-6 truncating mutants augment the cell motility in A549 cells. Quantification of a scratch wound-healing assay is presented. Values in the chart represent mean ± SD of length of wounds at the indicated time points. For statistical analysis, the wound length at each time point for a given truncation was compared to Td-Tom expressing cells (n = 12, p-value *<0.0005, **<0.00005 and ***<0.000005, unpaired t-test). See *Figure 2—figure supplement 1A* for representative images of wounds closure. (G) The chart represents the quantification of a trans-well matrigel cell invasion assay. Each bar is the mean ± SD from 6 independent experiments (p-value *<0.005, unpaired t-test). (H) C57BL/6J mice were intravenously injected (tail vein) with B16-F1 melanoma cells ectopically expressing the indicated constructs. After 14 days, the lungs were dissected and the number of melanoma colonies in lung were quantified. The upper and lower panels illustrate representative dorsal (D) and ventral (V) images of the lungs. See *Figure 2—figure supplement 2* for expression of truncations and *Figure 2—figure supplement 4* for the histological analysis. (I) The chart represents the number of melanoma colonies in lung in different p53-truncation expressing cells. Each bar is the average number of melanoma colonies in the lungs of individual mice with data pooled from three independent experiments (Mean ± SD, p-value *<0.005).

The following figure supplements are available for figure 2:

**Figure supplement 1.** p53 exon-6 truncations increase cell migration.

**Figure supplement 2.** Lentiviral ectopic expression of different p53 truncations.

**Figure supplement 3.** p53 exon-6 truncations increase expression of mesenchymal marker.

**Figure supplement 4.** p53 exon-6 truncation increases the colonization of melanoma cells in the lungs.

A549 cell line (lung cancer-derived epithelial line) induced changes in the morphological appearances of cells (*Figure 2D*) and the acquisition of mesenchymal-like features such as (i) the transition of filamentous actin from a cortical distribution to stress fibers formation (*Figure 2D*), (ii) decreased expression and localization of E-cadherin (*Figure 2D and E*), (iii) increased expression of vimentin, and (iv) increased expression of the master regulators of epithelial to mesenchymal transition (EMT): Slug, Snail and Zeb1 (*Figure 2E*) (*Lamouille et al., 2014*).

Consistent with an EMT-like phenotype, cells expressing p53 exon-6 truncations were also characterized by increased motility (*Figure 2F* and *Figure 2—figure supplement 1*) and extra-cellular matrix invasion (*Figure 2G*) (*Lamouille et al., 2014*).

As these features are typically associated with metastatic spread, we next employed a melanoma model to compare the lung colonization potential of diverse p53 truncating mutants. Specifically, we injected B16-F1 melanoma murine cells ectopically expressing the p53 W146*, R213*, G325* mutants, as well as the vector control Td-Tomato (*Figure 2—figure supplement 2*), into C57BL/6J mice via tail vein as previously described (*Overwijk and Restifo, 2001*). As shown in *Figure 2H and I* and *Figure 2—figure supplement 4*, at day 14 post-injection, we observed a dramatic increase in the number of melanoma colonies in lung in the case of cells expressing the p53 R213* mutant.

## *TP53* exon-6 truncating mutations are expressed in and required for cancer cell survival

In all eukaryotes, mRNA transcripts that contain premature stop codons might be detected and degraded by a surveillance pathway known as nonsense-mediated mRNA decay (NMD) (*Bateman et al., 2003*; *Behm-Ansmant and Izaurralde, 2006*). Yet, it has been shown that not all premature truncating transcripts undergo NMD, and that variation in NMD efficiency among different tissues, cell types and even individuals could lead to the expression of variable amounts of truncated proteins that could impact the clinical presentation and outcome of diseases. For example, no NMD-mediated mRNA diminution was observed in the lymphoblasts and bone cells of patients carrying premature termination codons in the collagen X gene (*Bateman et al., 2003*).

As NMD could impair the potential activity of *TP53* exon-6 truncating mutations, we compared the expression of *TP53* exon-6 truncating mutations and p53-WT in multiple tumor samples and tumor-derived cell lines.

Analysis of four individual sequencing studies done by the TCGA project indicated a large distribution in the expression of all p53 mRNAs and a comparable, though slightly decreased, expression of RNA transcripts harboring *TP53* truncating mutations (*Figure 3A*, *Figure 3—figure supplement 1* and *Supplementary file 2*).

We confirmed that *TP53* exon-6 truncating mutations partially escape NMD by western blot analysis of protein extracts from multiple tumor-derived cell lines harboring different *TP53* mutations (*Figure 3B*). As shown in *Figure 3C* and in *Figure 3—figure supplement 3*, we observed that p53 exon-6 truncating mutants were indeed expressed in the SW684, Calu-6 and DMS114 cell extracts. Notably, analyses of p53 expression upon knockdown with two independent p53 small hairpin RNAs (shRNA) verified that the bands we detected by western blot analysis corresponded to distinct p53 mutant forms (*Figure 3C* and *Figure 3—figure supplement 3A*).

Our studies based on the ectopic expression of p53 exon-6 truncating mutants indicated a possible gain of function activity of these p53 truncated isoforms (*Figure 2*). Having previously shown that these mutants are expressed in cancer cells, we next extended our functional studies to include cancer cell lines harboring *TP53* exon-6 truncating mutations.

Acute inactivation of p53 (i.e., 2–4 days) with two independent p53 shRNAs in cells that exclusively expressed the p53 R213* and R196* mutants resulted in a down-regulation of EMT markers and up-regulation of E-cadherin (*Figure 3D*, and *Figure 3—figure supplement 4A*). This was consistent with the pro-metastatic activities we observed in cells ectopically expressing *TP53* exon-6 truncating mutations and with the genetic data summarized in *Figure 1* and *Figure 2*

Yet, prolonged inactivation (i.e., more than 6 days) of p53 in tumor-derived cell lines harboring *TP53* exon-6 truncating mutations resulted in a dramatic decrease in the viability of the cells over time (*Figure 3E and F*). We confirmed that the decreased number of cells we found in our viability assay was due to an increase in cell death by measuring levels of expression of the apoptotic marker cleaved-PARP upon p53 knockdown (*Figure 3—figure supplement 6*). This appeared to be unique to cells expressing the R213* and R196* exon-6 truncating mutations (SW684, DMS114 and Calu-6) as silencing p53 in cell lines harboring (i) a wild type p53 allele (A549 and MCF7 cells), (ii) a 'hotspot' missense p53 mutant (AU565), (iii) a longer truncation (HCC1937), (iv) a shorter truncation (H2126), or (v) a p53 homozygous deletion (H1299) did not affect the number of viable cells (*Figure 3F and G*).

To eliminate the possibility that these differences were due to the efficiency of p53 inactivation or variance in the cells' sturdiness, we next measured p53 knockdown efficiency as well as the effect of inactivation of the essential gene *RPA3* on cell viability across all the cell lines (*Figure 3F and G* and *Figure 3—figure supplement 5*). In all cases we found similar efficiency of knockdown and comparable decrease in the viability of cells upon RPA3 silencing.

To provide further validation to our findings, we also employed an inducible CRISPR-Cas9 system to inactivate p53 in in-vitro and in-vivo model systems (*Senturk et al., 2016*) (*Figure 3H and I* and *Figure 3—figure supplement 7*). In these cases, the inactivation of p53 also resulted in a substantial decrease in cell viability only in those cells expressing *TP53* exon-6 truncating mutations.

## p53 exon-6 truncated mutants partially localize to the mitochondria and regulate mitochondrial transition pore permeability by interaction with cyclophilin D

p53-psi is a naturally occurring alternative splice isoform generated by the use of an alternative cryptic splicing acceptor site in intron-6 (*Figure 4—figure supplement 1*) with an approximate molecular weight of 35 kDa. As demonstrated in Senturk et al., a p53-psi like protein can also be generated by mutations occurring at the splice acceptor site in exon-7 (Hop62, c.673–2A>G *TP53* mutation).

From a molecular and phenotypic standpoint, the products of *TP53* exon-6 truncating mutations highly resemble p53-psi (*Figure 4—figure supplement 1*). Like p53-psi, these p53 mutants lack most of the domains required for p53 canonical tumor suppressor activities (i.e., nuclear localization, oligomerization domains, and the alpha helix required for p53-DNA binding) and are capable of reprogramming the cells towards the acquisition of pro-metastatic features. Therefore,

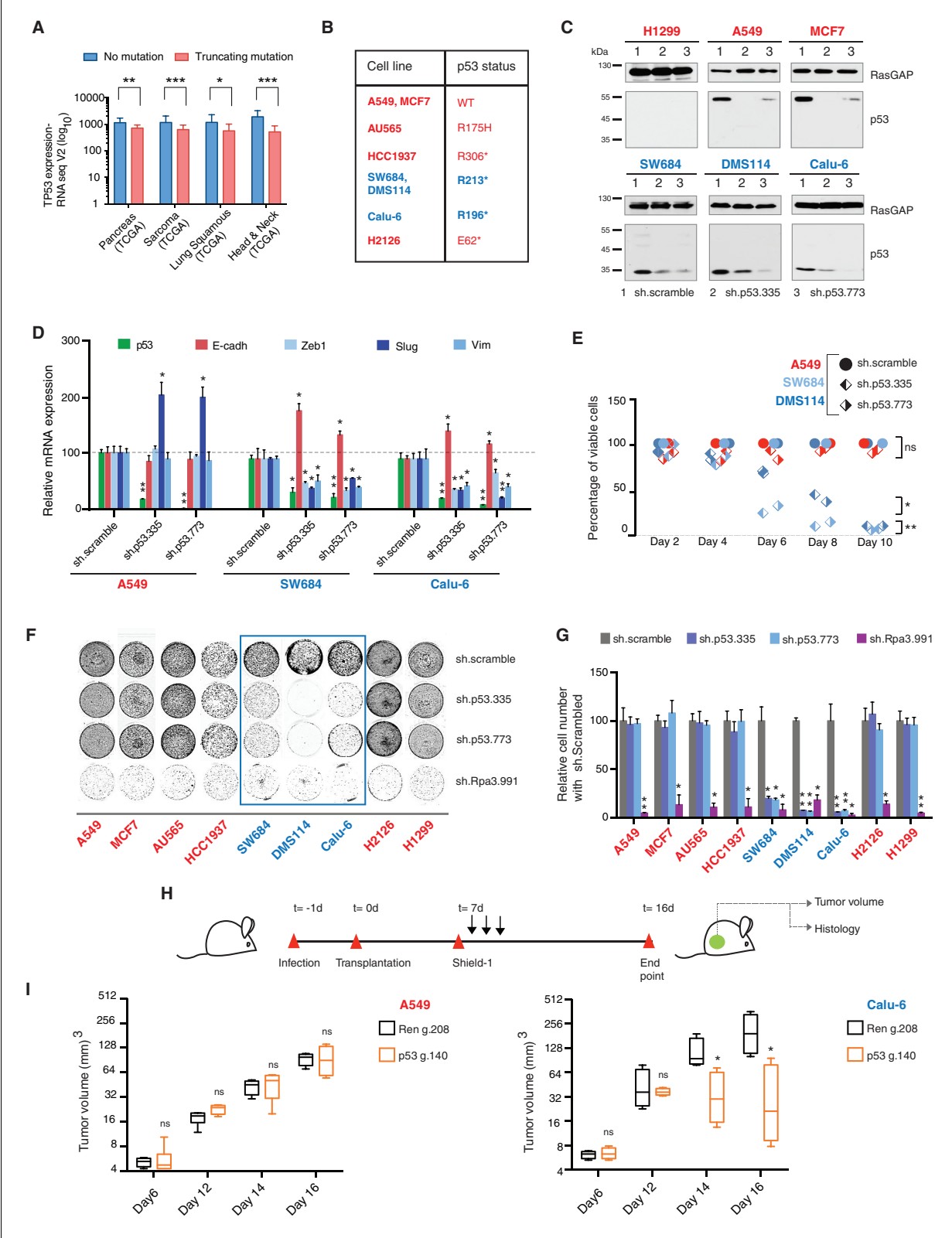

**Figure 3.** *TP53* exon-6 truncating mutations are required for EMT and cell survival. (A) The chart represents the median distribution of mRNA transcripts from the indicated tumors with p53 truncating mutation (Red) and no mutation (Blue), based on TCGA datasets. See *Figure 3—figure supplement 1* and *Supplementary file 2* for further details. (B) The table summarizes *TP53* mutation status in the cell lines utilized in this study. (C) Western blot analysis of the indicated cell lines using a p53 N-terminal specific (DO1) antibody and an antibody against RasGAP as loading control. *Figure 3 continued on next page*

*Figure 3 continued*

Quantification is provided in *Figure 3—figure supplement 3B*. (D) The chart represents themRNA expression analysis of the indicated genes in A549, SW684 and Calu-6 cell lines. mRNA expression was quantified by SYBR-green-based RT-qPCR. Each bar is the average of 3 replicates and represents the mRNA expression of the indicated gene relative to GAPDH (p-value *<0.05 and **<0.005, unpaired t-test). Analysis of additional cell lines is provided in *Figure 3—figure supplement 4A* (E) Each dot represents the percentage of viable cells compared to scramble shRNA in cells expressing p53-WT (A549) or exon-6 truncating mutations (SW684 and DMS114) upon knockdown of p53 with two independent shRNA constructs. Each dot represents the mean of 9 individual replicates. Efficiency of knock down is provided in *Figure 3—figure supplement 4B* (F) Crystal violet staining of the indicated cell lines upon p53 knockdown with two independent p53 shRNAs. A scramble shRNA was used as a negative control while shRNA targeting the essential gene *RPA3* was used as a positive control. The quantification of knockdown efficiency is provided in *Figure 3—figure supplement 5* (G) The chart depicts the percentage of viable cells 8 days post infection with the indicated shRNA constructs relative to scramble shRNA control. Each bar represents the mean of 9 individual replicates (p-value *<0.005, **<0.0005, unpaired t-test). See *Figure 3—figure supplement 5* for knockdown efficiency. See *Supplementary file 4* for shRNA sequences. (H) Workflow of the transplantable model system used in this study. A549 (p53-WT) and Calu-6 (p53 R196*) cells were transduced with lentivirus constructs expressing an inducible CRISPR-Cas9 (DD-Cas9) targeting p53 (p53 g.140) and as negative control targeting Renila (Ren.g.208). Cells were transplanted sub-cutaneously in immune-deficient mice. When the tumors reached an approximate size of 4–5 mm in diameter; mice were treated with Shield-1 (1 μg). Tumor volume was determined at the indicated time points. See *Supplementary file 5* for sgRNA sequences. (I) The charts illustrate quantification of tumor volumes (mean ± SD) in the indicated cohorts at given time points (n = 4, p-value *<0.05, unpaired t-test). Validation of p53 inactivation is provided in *Figure 3—figure supplement 7B and C*.

The following figure supplements are available for figure 3:

**Figure supplement 1.** The P53 mRNA expression is comparable between p53-WT and p53 truncations.
**Figure supplement 2.** Schematic of the p53-WT or truncated forms expressed in the cell lines utilized in this study.
**Figure supplement 3.** Validation of p53 expression by immuno-blot and efficient knockdown with indicated p53 specific shRNA.
**Figure supplement 4.** Differential expression of indicated genes and knockdown efficiency of indicated p53 shRNAs.
**Figure supplement 5.** Knockdown efficiency of indicated shRNAs in following cell lines.
**Figure supplement 6.** Knockdown of p53 induces apoptosis in cell expressing in p53 exon-6 truncation.
**Figure supplement 7.** Inactivation of p53 by using CRISPR-Cas9 decreases cell survival of p53 exon-6 truncation expressing cells.

---

unsurprisingly, we not only found that p53 exon-6 truncating mutants were excluded from the nucleus (*Figure 4B*), but also that they lacked transcriptional activities (*Figure 4C*).

Previous studies have indicated that p53-WT under stress conditions could localize to the mitochondria and bind to the mitochondria permeability transition pore (MPTP) regulator Cyclophilin D (CypD) through a domain present from amino acid 80 to amino acid 220 of the p53 protein (*Vaseva et al., 2012*). More recently Senturk et al., showed that p53-psi is constitutively localized to the mitochondria where it also binds to CypD. Specifically, they were able to demonstrate that the tumor promoting activities of p53-psi requires its molecular and functional interaction with CypD (*Senturk et al., 2014*) (*Figure 4A*).

CypD is a peptidyl-prolyl isomerase and the only validated regulatory component of the MPTP (*Baines et al., 2005*; *Schinzel et al., 2005*; *Giorgio et al., 2010*). CypD activity can be pharmacologically inactivated by Cyclosporin A (CsA). CsA was initially isolated from the fungus Tolypocladium inflatum (*Heusler and Pletscher, 2001*). It is best known as an immunosuppressant drug that reduces the activity of the immune system by interfering with the activity and growth of T cells (*Bunjes et al., 1981*). In addition to its effect on T cells, CsA has also been shown to be a potent inhibitor of CypD by preventing the binding of CypD to other components of the MPTP (*Halestrap and Davidson, 1990*, *Nicolli et al., 1996*; *Baines et al., 2005*).

As the domains that are required for p53 mitochondria localization and its interaction with CypD are conserved in *TP53* exon-6 truncating mutations, we examined the sub-cellular distributions of different p53 truncating mutants and their interactions with CypD. We found that p53 exon-6 truncating mutants were partially localized to the mitochondria (*Figure 4D* and *Figure 4—figure supplement 2*) and, as shown by our immunoprecipitation experiments, could bind to CypD in the

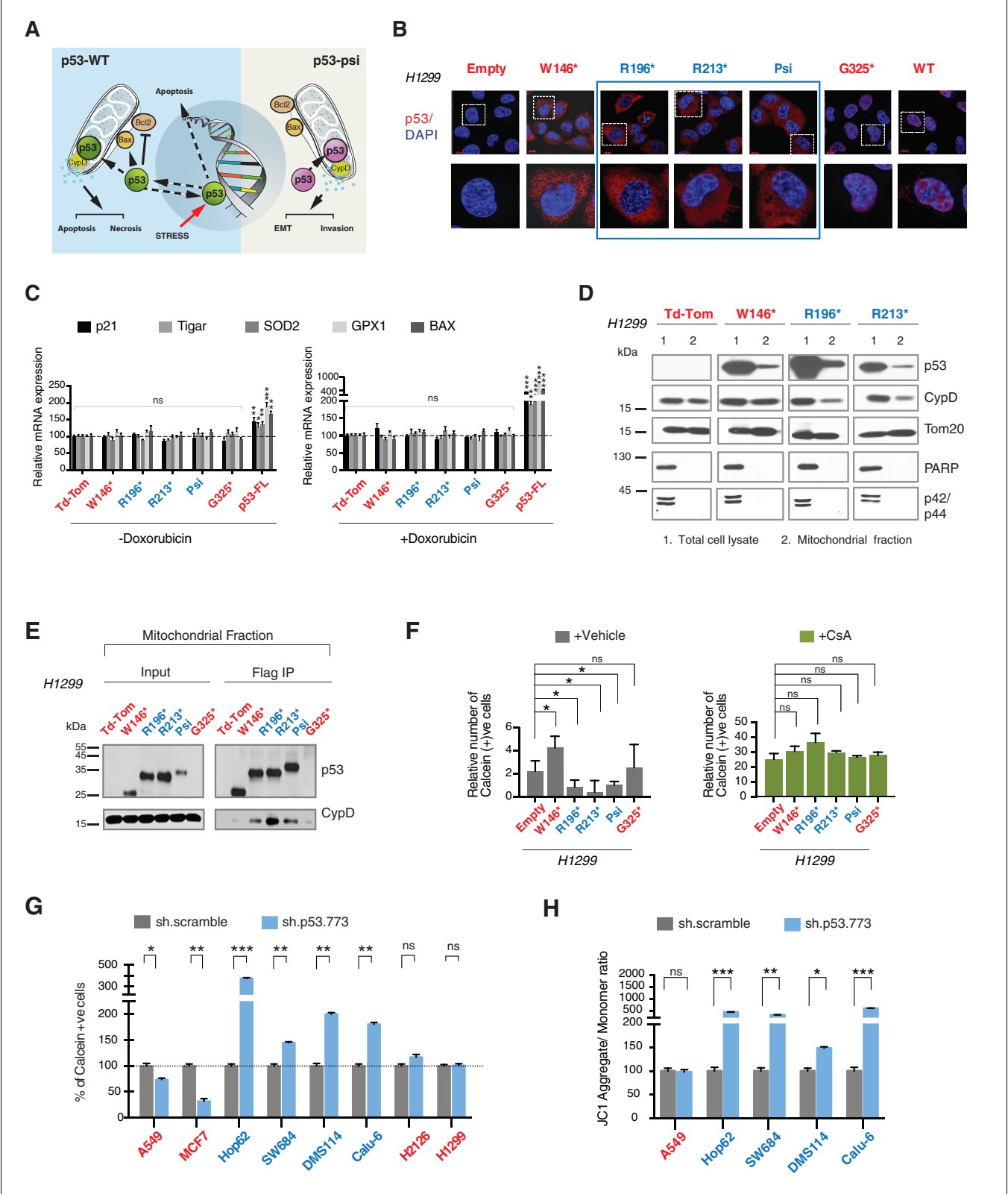

**Figure 4.** CypD activity is required for phenotypes associated with *TP53* exon-6 truncating mutations. (**A**) Schematic of p53-psi activities as reported by Senturk et al., unlike from p53-WT, p53-psi does not localize in the nucleus and does not have transcriptional capabilities. Yet, p53-psi can translocate to the mitochondria where it binds to CypD and via modification of the inner pore permeability induces pro-metastatic features. (**B**) p53 truncating mutants are excluded form the nucleus. Immuno-staining of H1299 cells with the p53 N-terminal specific (DO1) antibody (red). DAPI (blue) is used as

*Figure 4 continued on next page*

*Figure 4 continued*

counterstain. (C) The chart represents an expression of p53 target genes upon ectopic expression of different p53 truncations in A549 cells either in the absence or presence of Doxorubicin (1 µM) for 24 hr. mRNA expression was quantified by SYBR-green based RT-qPCR. Each bar is the average of 3 replicates and represents the mRNA expression of the indicated genes relative to GAPDH (p-value, *<0.05, **<0.005 and ***<0.0005 unpaired t-test). (D) p53 truncating mutants are partially localized in the mitochondria. Western blot analysis of total cell extracts and of mitochondrial fractions shows translocation of p53 truncations in mitochondria. Purity of fractions was verified with antibodies specific for the mitochondria matrix protein CypD, mitochondria outer-membrane associated protein Tom20, nuclear protein PARP and cytoplasmic protein p42/44. (E) Immuno-precipitation analysis of the mitochondrial fraction from H1299 cells expressing FLAG-tagged different p53 truncations. Cell extracts were immuno-precipitated with a FLAG specific antibody and analyzed by western blot with a p53 N-terminal specific (DO1) antibody and CypD specific antibody. (F) p53 truncating mutants increase the mitochondria inner pore permeability. See Materials and methods for details on assay design. The graph indicates the relative number of H1299 cells expressing different p53 truncations that retain calcein fluorescence in mitochondria, upon CoCl$_2$ treatment, in the presence or absence of CsA (2 µM) (n = 3, p-value *<0.05, unpaired t-test). (G) The graph indicates the percentage of cells retaining calcein fluorescence in mitochondria upon CoCl$_2$ treatment in the indicated cell lines upon p53 knockdown with shRNA. See *Figure 4—figure supplement 3* for p53 knockdown efficiency. Note, there is an increase in the number of calcein positive cells (decreased permeability) upon p53 knockdown in cells harboring p53-psi or p53 exon-6 truncating mutations (n = 3, p-value *<0.05, **<0.005 and ***<0.0005, unpaired t-test). (H) The chart indicates the ratio of JC-1 aggregate relative to monomer in the indicated cells after p53 knockdown with a p53 shRNA lentiviral construct relative to scrambled shRNA. Note the increase in the number of J aggregates (increased mitochondrial polarization) upon p53 knockdown in cells harboring p53-psi or p53 exon-6 truncating mutations (n = 3, p-value *<0.0005, **<0.00005 and ***<0.000005, unpaired t-test).

The following figure supplements are available for figure 4:

**Figure supplement 1.** p53-psi molecularly resembled *TP53* exon-6 truncating mutations.

**Figure supplement 2.** p53-psi and p53 exon-6 truncations localize to mitochondria without affecting expression and localization of CypD.

**Figure supplement 3.** Knockdown efficiency upon infection with shRNA targeting p53.

**Figure supplement 4.** p53-psi and p53 exon-6 truncations regulate MPTP and mitochondrial polarization in CypD dependent manner.

**Figure supplement 5.** Knockdown efficiency upon infection with shRNA targeting p53.

**Figure supplement 6.** Knockdown of p53 exon-6 truncations and p53-psi does not affect the expression and localization of CypD.

**Figure supplement 7.** Knockdown of CypD does not affect the expression and localization of mutant p53 isoforms.

mitochondrial fractions (*Figure 4E*). The binding of p53 exon-6 truncating mutants with CypD was not due to changes in the expression of CypD, as neither the ectopic expression of *TP53* exon-6 truncating mutations nor decreasing their expression had any effects on CypD expression or its mitochondrial localization (*Figure 4—figure supplement 2* and *6*). Similarly, knockdown of CypD in cells expressing p53-psi or exon-6 truncating mutants did not affect the localization of truncated p53 isoforms to mitochondria (*Figure 4—figure supplement 7*).

To test the functional role of a CypD/p53 exon-6 truncations interactions, we analyzed changes in the permeability of the MPTP by using a calcein fluorescence assay in cells ectopically expressing W146*, R196*, R213*, p53-psi and G325*. We found an increased permeability of the MPTP only in the case of p53-psi and the R196* and R213* mutants. Interestingly, in the cells expressing the W146* p53 mutants we instead observed a decrease of MPTP permeability (*Figure 4F*).

To confirm that these changes in mitochondrial permeability were dependent on CypD, we then inhibited CypD activity using CsA. We found that in the presence of CsA, the increase in the pore permeability that we observed in cells expressing R196*, R213* p53 exon-6 mutants and p53-psi was completely ablated (*Figure 4F*). To further validate the possible function of p53 exon-6 truncations and p53-psi in regulating the MPTP function, we performed similar experiments in tumor-derived cell lines with p53-psi splicing mutation (Hop62), exon-6 truncating mutation (SW684, DMS114 and Calu-6), p53-WT (A549, MCF7), a shorter truncation (H2126) and a p53 homozygous deletion (H1299) upon knock down of p53. As shown in *Figure 4G*, only cells expressing p53-psi and exon-6 truncating mutations showed a decrease in mitochondrial permeability after the knockdown

of p53. Similarly, CypD silencing resulted in a decrease in mitochondrial permeability only in those cells expressing p53-psi and exon-6 truncating mutations (*Figure 4—figure supplement 4A and C*).

An increase in the mitochondria pore permeability is predicted to reduce the mitochondrial polarization. Hence, we also performed a JC-1 based assay to analyze any changes in the mitochondrial polarization upon inactivation of p53 and CypD. As shown in *Figure 4H* and *Figure 4—figure supplement 4B*, knockdown of p53 or CypD in cell lines expressing p53-psi and exon-6 truncating mutations specifically resulted in the hyperpolarization of mitochondria.

## Cyclophilin D activity is required for phenotypes associated with *TP53* exon-6 truncating mutations

Having shown that p53 exon-6 truncating mutants and p53-psi regulate the MPTP in a CypD dependent fashion (*Figure 4*), we next conducted studies aimed at understanding a functional role of the inner pore regulator CypD in mediating the acquisition of pro-metastatic features and survival in cells expressing p53-psi or p53 exon-6 truncating mutants. To this end, we genetically reduced the expression of CypD either by shRNA mediated knockdown in-vitro or by CRISPR-Cas9 mediated gene editing in a mouse model system.

*Figure 5A* demonstrates that short-term inactivation of CypD led to a decrease in mesenchymal markers in cells specifically expressing p53-psi or exon-6 truncating mutants (Hop62, DMS114 and Calu-6). When p53 was inactivated in these cells, we observed a reduction in cell viability; analogously, CypD long-term knockdown also led to a reduction in viable cells in in-vitro cell viability experiments (*Figure 5C and D* and *Figure 5—figure supplement 1B*). This result was also recapitulated by treatment of cells with the novel CypD inhibitor, C-9 (*Figure 5B*) (*Valasani et al., 2016*).

To provide evidence that the dependency of cells expressing p53 exon-6 truncating mutants on CypD expression was not only restricted to in-vitro settings, we next extended our studies to an in-vivo model system based on sub-cutaneous transplantation of tumor cell lines in immune-compromised mice. *Figure 5E and F* show that the inactivation of CypD expression in xenograft models using an inducible CRISPR-Cas9 gene editing system decreased the tumor volume of p53 exon-6 truncating mutant-expressing cells (Calu6), but not the tumor volume of p53-WT expressing cells (A549) (*Figure 5—figure supplement 2*).

## Discussion

In summary, our analysis of human tumors, combined with the detailed molecular characterization of *TP53* exon-6 truncating mutations, offers strong support for the idea that chromosome 17p deletions and, particularly, *TP53* mutations produce a multiplicity of alleles with diverse activities that contribute differently to tumorigenesis by providing distinct, selective advantages (*Petitjean et al., 2007*; *Brosh and Rotter, 2009*; *Olivier et al., 2010*; *Oren and Rotter, 2010*; *Liu et al., 2016*). Our studies, in fact, revealed that p53 exon-6 truncating mutants not only lack transcriptional activities and the capacity to respond to DNA damage, but are also uniquely able to activate a pro-tumorigenic cellular program.

Consistent with the capabilities of these p53 mutants to promote rather than halt tumorigenesis, *TP53* exon-6 truncating mutations are highly abundant and are enriched in certain tumors (*Figure 1*). Although, in principle this implies that these specific mutants could increase the fitness of tumors, we cannot exclude the possibility that the higher than expected frequency we observed in tumors could alternatively be explained by a specific etiology and/or a particular mutagenic modality. Additional experiments in mouse model systems will be required to better understand the ontogeny of these mutations.

One interesting feature of the p53 exon-6 truncating mutants is their similarity to the naturally occurring p53 isoform p53-psi. Much like p53-psi, they lack part of the DNA binding and oligomerization domains as well as the nuclear localization sequences; however, they are partially localized to the mitochondria where they are able to bind and activate Cyclophilin D.

CypD is a positive regulator of the opening of the MPTP. Inhibition of CypD activity, either genetically or pharmacologically, has been reported to reduce mitochondrial permeability and induce changes in mitochondrial membrane potential. Hence, unsurprisingly, we observed that the expression of *TP53* exon-6 truncating mutations is sufficient to increase the MPTP.

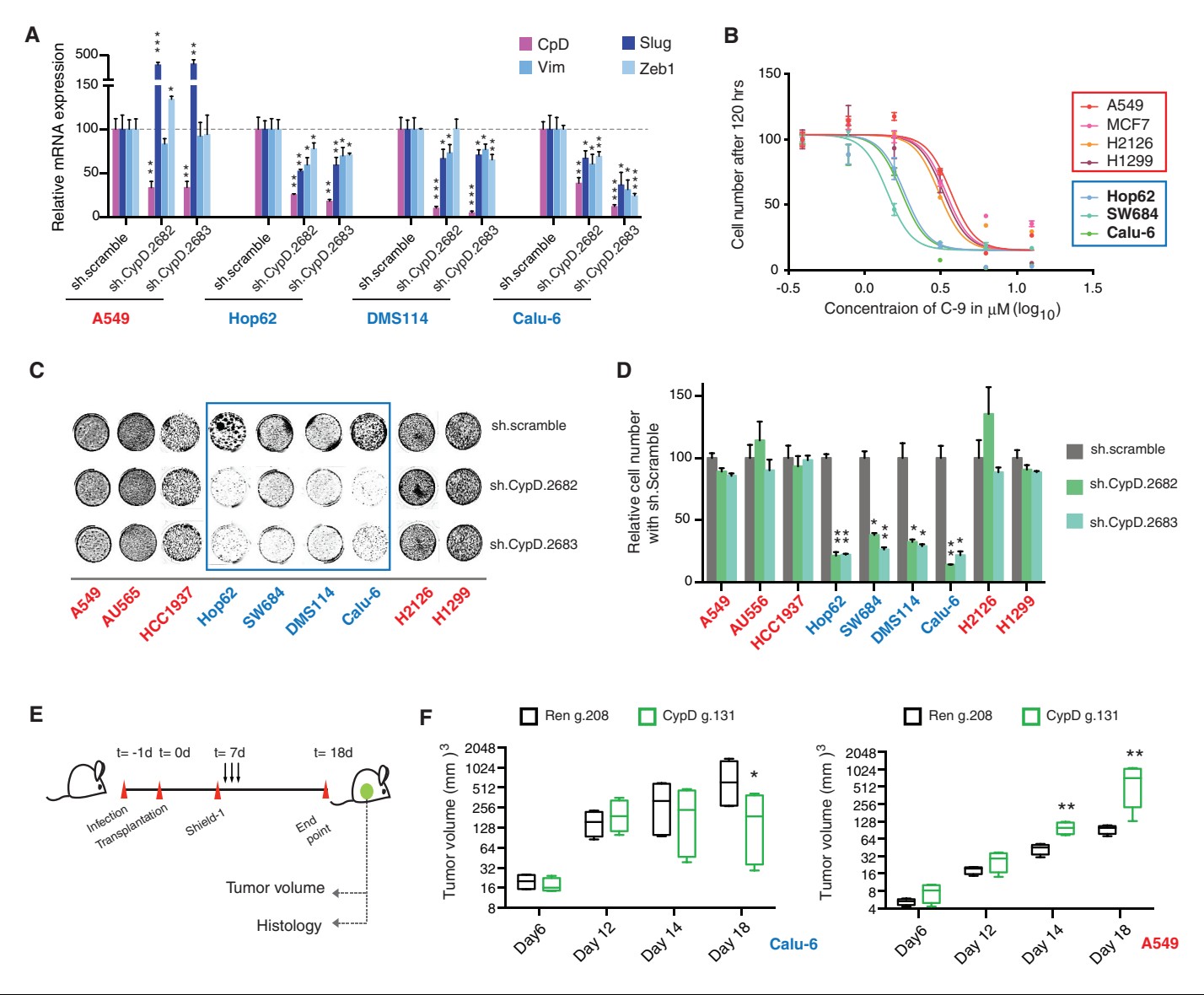

**Figure 5.** CypD activity is required for phenotypes associated with *TP53* exon-6 truncating mutations. **(A)** CypD is required for maintaining cells in a mesenchymal like state. The chart represents mRNA expression analysis of the indicated genes in A549 (p53-WT), Hop62 (p53-psi), DMS114 (p53-R213*) and Calu-6 (R196*) cell lines after CypD knockdown. Cells harboring p53-psi or *TP53* exon-6 nonsense mutations are indicated in blue. mRNA expression was quantified by SYBR-green-based RT-qPCR. Each bar is the average of 3 replicates and represents mRNA expression of the indicated gene relative to GAPDH (p-value, *<0.05, **<0.005 and ***<0.0005, unpaired t-test). See *Figure 5—figure supplement 1A* for analysis in additional cell lines. **(B)** CypD is required for the survival of cells harboring p53-psi splice or *TP53* exon-6 truncating mutations. The graph represents cell survival curve of indicated cell lines when treated with CypD inhibitor C-9 for 120 hr. **(C)** Crystal violet staining of the indicated cell lines upon CypD knockdown with two independent shRNAs. A scramble shRNA was used as negative control. The quantification of knockdown efficiency is provided in *Figure 5—figure supplement 1B*. **(D)** The chart depicts the percentage of viable cells 8 days after infection with the indicated CypD shRNA constructs relative to scramble shRNA control. Each bar represents the mean of 9 individual replicates (p-value *<0.0005 and **<0.00005, unpaired t-test). **(E)** Workflow of the transplantable model system used in this study. A549 (p53-WT) and Calu-6 (p53 R196*) cells were transduced with an inducible CRISPR-Cas9 (DD-Cas9) targeting CypD (CypD g.131) and Renila (Ren g.208). Cells were transplanted sub-cutaneously in immune-deficient mice. When the tumors reached an approximate size of 4–5 mm in diameter, mice were treated with Shield-1 (1 µg). Tumor volume was determined at the indicated time points. See *Supplementary file 5* for sgRNA sequences. **(F)** The charts illustrate quantification of tumor volumes (mean ± SD) in the indicated cohorts at given time points (n = 4, p-value *<0.05, unpaired t-test). Validation of CypD inactivation is provided in *Figure 5—figure supplement 2*.

The following figure supplements are available for figure 5:

**Figure supplement 1.** Differential expression of indicated genes and knockdown efficiency of CypD shRNAs.

*Figure 5 continued on next page*

*Figure 5 continued*

**Figure supplement 2.** Inactivation of CypD by CRISPR-Cas9 in in vitro and in vivo systems.

In the past, an augmented mitochondrial permeability has always been associated with decreased cell viability. Yet, cells expressing *TP53* exon-6 truncating mutations are proliferating under normal conditions and are actually dependent on CypD for their survival. These observations are consistent with more recent reports indicating that the MPTP has a role not only in inducing cell death under cellular stress, but also in physiological processes (*Kwong and Molkentin, 2015*) and in the induction of phenotypic changes associated with pro-metastatic features (*Senturk et al., 2014*).

The requirement of mitochondria localization, interaction with CypD and the lack of transcriptional activities distinguish *TP53* exon-6 truncating mutations from other p53 gain of function mutants with pro-tumorigenic functions (e.g. R175, G245, R248, R249, R273 and R282). Despite the fact that p53 gain of function mutants are also capable of inducing pro-metastatic features, we found that their activities strongly depend on their nuclear localization and transcriptional activities.

In our study, structure-function analysis has also shown that the p53 W146* mutant is unable to interact with CypD despite its ability to localize to the mitochondria. Interestingly and contrary to p53-psi and p53 exon-6 truncating mutants, the expression of the p53 W146* mutant increased the expression of E-cadherin instead of reducing it (*Figure 2D and E*), and decreased the permeability of the MPTP in a calcein assay (*Figure 4F*). To explain this observation, it is tempting to speculate that the p53 W146* mutant may interact with other components of the MPTP and antagonize the function of CypD. Further studies will be required to illuminate the underpinning mechanisms.

The selective dependencies of cancer cells harboring exon-6 *TP53* truncating mutations to CypD activity (*Figure 5B–F*) is particularly exciting, as it begs the design of novel targeted therapeutics. Notably, the high prevalence of p53-psi or *TP53* exon-6 truncating mutations in certain tumors also suggests that this class of *TP53* mutations represents a strong precision medicine candidate target comparable to the well-documented EGFR, ROS and ALK mutations in NSCLC (*Korpanty et al., 2014*).

In conclusion, our studies indicated that multiple *TP53* exon-6 truncating mutations, despite lacking transcriptional and canonical p53 tumor suppressor activities, can reprogram the cells' signaling networks, change the 'dependency' of cells and their cell state. As these mutations structurally and functionally mimic the naturally occurring p53-psi isoform, we propose that *TP53* exon-6 mutations are best described as 'separation of function' rather than simply 'gain of function' or 'loss of function' Interestingly, approximately one-third of all human genetic disorders are caused by mutations that generate premature stop codons (*Frischmeyer and Dietz, 1999*). Hence, *TP53* exon-6 truncating mutations could represent a paradigm for other diseases that could similarly be driven by the separation of function isoforms.

## Materials and methods

### Animals

#### Xenograft assay

All animal experiments were performed in accordance with the National Research Council's Guide for the Care and Use of Laboratory Animals. Protocols were approved by the Cold Spring Harbor Laboratory Animal Care and Use Committee. Female NU/NU mice 6-weeks old were purchased from Charles River (Wilmington, Massachusetts). A549 and Calu-6 lung cancer cells were plated and infected in-vitro with lentiviruses carrying Renila and p53 sgRNAs at a multiplicity of infection (MOI) of 1. Xenograft tumors of A549 and Calu-6 cells with inducible-cas9 expression were established by subcutaneous injection of $5 \times 10^5$ cells in 100 µL volume mixed with 1:1 dilution basement membrane matrix with biological activity (Matrigel, BD Biosciences, San Jose, California). Three to four animals per group were used in each experiment. When tumors reached a palpable size, animals were administered with 1 µg peritumoral injection of Shield-1 (diluted in 100 µL PBS), once per day for the duration of four days. Tumor growth was followed for two weeks using a vernier caliper

(volume = ((ds*short*)$^2$ × (d*long*))/2). At the end of the experiment, the mice were sacrificed. Tumors were extracted and fixed in freshly prepared 4% paraformaldehyde for 24 hr.

## Tail vein assay

Tail vein assay for B16-F1 cells with p53 truncations was conducted as previously described (*Overwijk and Restifo, 2001*). B16-F1 cells were collected at 50% confluency and a final suspension of cells with 4 × 10$^5$ cells/mL was prepared in HBSS buffer. C57BL6 mice were injected with 0.5 mL of cell suspension intravenously in the tail vein. Since B16-F1 melanoma have inherent colonization property, the mice were sacrificed at day 14 post-injection. We changed the end-point of experiment to day 14 instead of 18 to avoid saturation of lung with melanoma colonies which would had underestimated the increase in colonization potential of specific clone. For quantification of lung colonies, extracted lungs were fixed in freshly prepared 4% paraformaldehyde for 24 hr. After fixation, lungs were embedded in agarose, sectioned vertically and transferred to slides for H and E staining.

## Cell lines

All cell lines were obtained from the American Type Culture Collection (ATCC, Manassas, Virginina), except Hop62 which was obtained from the National Cancer Institute (NCI) with provided information of authenticity in the year 2015. All of the cell lines were regularly monitored for mycoplasma contamination by using Lonza mycoalert mycoplasma detection kit as per the manufacturer's instructions. All the cell lines tested negative for Mycoplasma contamination. A549 (RRID: CVCL_0023), AU565 (RRID: CVCL_1074), H1299 (RRID: CVCL_0060), HCC1937 (RRID: CVCL_0290) and Hop62 (RRID: CVCL_1285) cells were cultured in RPMI supplemented with 10% Fetal Bovine Serum (FBS, HyClone), Penicillin-Streptomycin (10,000 units/mL, Gibco). B16-F1 (RRID: CVCL_0158), Calu-6 (RRID: CVCL_0236), H2126 (RRID: CVCL_1532), HEK-293T (RRID: CVCL_0063), MCF7 (RRID: CVCL_0031), and SW684 (RRID: CVCL_1726) cells were cultured in DMEM supplemented with 10% Fetal Bovine Serum (FBS, HyClone), Penicillin-Streptomycin (10,000 units/mL, Gibco). DMS114 (RRID: CVCL_1174) cells were cultured in Waymouth media supplemented with 10% Fetal Bovine Serum (FBS, HyClone), Penicillin-Streptomycin (10,000 units/mL, Gibco). All of the cell lines were incubated at 37°C with 5% CO$_2$ incubation.

## Generation of cell lines

For the constitutive ectopic expression of p53-WT, p53-psi and other p53 truncated forms, we used a lentiviral gene expression system. cDNAs encoding p53-WT, p53-psi and different p53 truncations were cloned into the pENTR4 (Invitrogen, Carlsbad, California) vector by using A549 cells cDNA as a template for *TP53*. Using the Gateway technology, each pENTR4 vector was recombined with pENTR5 vector and pLenti6.4 destination vector. For our experiment, we used pENTR5 vector that contains the human EF-1α promoter.

## Virus production

1. For pLenti6.4 System: HEK-293T cells were co-transfected with the constructs encoding the genes of interest with the packaging plasmids as LP1, LP2 and p-VSV-G using lipofectamine 2000 reagent (Life Technologies, Carlsbad, California). 10 mL of virus particles were collected after 48 hr of transfection by clarifying the supernatant through 0.45 μm filter membrane. Cells were infected and selected by blasticidin at concentration of 10 ug/mL in 3–5 days.
2. For PLKO.1 shRNA System: All shRNA constructs were obtained from Sigma-Aldrich (St. Louis, Missouri), except PLKO.1 scrambled and p53-773 that were obtained from Addgene. HEK-293T cells were co-transfected with the PLKO.1 constructs with the packaging plasmids as BH10, Rev and VSV-G using lipofectamine 2000 reagent. 10 mL of virus particles were collected after 48 hr of transfection by clarifying the supernatant through 0.45 μm filter membrane. The sequence of shRNAs used in this study is listed in *Supplementary file 4.*
3. For DD-Cas9 System: DD-Cas9 (Destabilized Cas9) system was designed in lab by (*Senturk et al., 2016*). sgRNAs were designed using an algorithm on http://crispr.mit.edu/. Virus packaging was achieved by transiently co-transfecting HEK-293T cells on 10 cm culture dish with the constructs encoding the sgRNAs of interest with DD-Cas9 along with the packaging plasmid psPAX2 and envelope plasmid pMD2.G (Didier Trono, Addgene) using lipofectamine 2000 reagent. 10 mL of viral particles were collected after 48 hr of transfection by

clarifying the supernatant through 0.45 µm filter membrane. Virus transduction was optimized in order to achieve low MOI transduction. Typically, 500–2000 µL virus particles from 10 mL stock were used to infect $1 \times 10^6$ cells on a 10 cm culture dish in 10 mL total volume of culture medium. Shield-1, obtained from Cheminpharma (Branford, connecticut), was solubilized in pure ethanol and was added to culture media with given concentrations. The sequence of oligonucleotides for cloning all sgRNAs used in this study is listed in *Supplementary file 5.*

## Western blot analysis

Protein samples were isolated by re-suspending cell pellets in RIPA buffer (50 mM Tris-HCl at pH 7.6, 150 mM NaCl, 1% NP-40, 0.5% Na deoxycholate, 0.1% SDS with Protease inhibitors). After removal of the debris, samples were quantified with colorimetric BCA kit (Pierce, Rockford, Illinois). For p53 expression 10 µg of protein extract for p53-WT expressing cells and ectopic expression cell lines or 50–75 µg of protein extract for p53 truncation expressing cells lines were electrophoresed on 6–12% gradient gels and wet-transferred to nitrocellulose membranes. For other proteins 20 µg of protein extract were electrophoresed on 6–12% gradient gels and wet-transferred to nitrocellulose membranes. After 1 hr blocking with 5% nonfat dry milk in 1X TBS, 0.1% Tween20 at room temperature, membranes were incubated with antibodies diluted in 1% w/v BSA as follows; p53-DO1 mouse mAb(1:1000, EMD Millipore, Billerica, Massachussetts, RRID: AB_213402), α-tubulin DM1A mouse mAb (1:50000, EMD Millipore, RRID: AB_11204167), Anti-Ras-GAP mouse mAb (1:1000, BD Biosciences, RRID: AB_397455), Tom 20 (FL145) rabbit polyclonal Ab (1:1000, Santa-Cruz Biotechnology, Santa Cruz, California,RRID: AB_2207533), PARP (46D11) Rabbit mAb (1:1000, Cell signaling technology, Danvers, Massachusetts, RRID: AB_10695538) cleaved PARP (Asp214) Rabbit Ab (1:1000, Cell signaling technology, RRID: AB_331426), p42/44 MAPK (ERK1/2) antibody (1:1000, Cell signaling technology, RRID: AB_330744) and CypD mouse mAB (1:5000, Abcam, Cambridge, Massachusetts, RRID: AB_10864110). All incubations were performed overnight at 4°C. Membranes were rinsed thoroughly with 1X TBS-T and then incubated with species-specific HRP-tagged secondary antibodies (1:10000, Bio-Rad, Hercules, California). Western blots were developed by incubating the membranes with Supersignal west femto maximum sensitivity substrate diluted in Pierce ECL western blotting substrate (1:10 v/v) solution (Thermo Fisher Scientific, Waltham, Massachusetts) for 4 min.

## RNA isolation and RT-qPCR

Cells were rinsed twice and harvested with ice cold PBS. Pellets were lysed in 800 µL Trizol (Invitrogen) and RNA was extracted according to the manufacturer's instructions. Contaminating DNA was removed by RNase-free DNase (Promega, Madison, Wisconsin) treatment for 30 min at 37°C. cDNA was prepared from 2 µg total RNA using ImProm-II Reverse Transcription System (Promega) with 16mer oligo(dT). RT-qPCR was performed using Power SYBR Green PCR Master Mix as per the manufacturer's instruction (Applied Biosystems, Carlsbad, California).

## Cell growth assay

For cell growth assay, an equal number (5000–15000 cells/well) of cells was plated in quadruplets in 12-well plates (BD falcon, Corning, New York) after infection with either shRNAs or DD-Cas9 virus particles. Cell growth was followed for 8–15 days, and media was changed every 3–4 days. The cells were split at intermitted time points to avoid reaching over-confluency. Cells were washed with PBS to get rid of the floating cells and fixed in 4% Formaldehyde in PBS (V/V) for 10–15 min. Fixed cells were stained by staining solution (0.1% Crystal violet in 10% ethanol) for 20 min. The staining solution was aspirated from the wells, and the cells were washed with water three times to get rid of any extra stain. Stained cells were air dried and imaged by using Licor Odyssey. To quantify the cell numbers, cells were destained by using 10% Acetic acid and absorbance of de-stained solution was measured at 590 nm at appropriate dilutions.

## Wound healing assay

To perform the wound-healing assay, cells were plated in a 6-well plate (BD-Falcon) and allowed to grow up to 80–90% confluency. A wound was introduced using a P200 tip, and was imaged at indicated time points using a Zeiss Observer Live Cell inverted fluorescence microscope. The

quantification of wound closure was performed by measuring the length of wound at indicated times by using ImageJ software.

## Invasion assay

The cell invasion assay was performed using a Cytoselect 24-well cell invasion assay Basement membrane kit as per manufacturer's instructions (Cell Biolabs Inc, San Diego, California).

## Immunofluorescence

Cells were grown on glass coverslips in 24-well cell culture plates and collected at appropriate confluency. Cells were fixed with 4% Para-formaldehyde and permeabilized in 0.1% Triton X-100 in PBS for 10 min. Fixed cells were washed three times in PBS and blocked with 1% BSA in PBS for 1 hr. After washing three times with PBS, cells were incubated with the primary antibody diluted in 1% BSA for overnight at 4°C. Immune complexes were then stained with the indicated secondary antibodies (Invitrogen). DAPI was used for nuclear staining. Stained cells were mounted with Vectashield mounting medium (Vector Laboratories, Burlingame, California) and analyzed under theconfocal microscope. The antibodies used for immunofluorescence were p53-DO1 mouse mAb (1:100, EMD Millipore) and E-Cadherin (1:200, BD Biosciences). Alexa Fluor 488-tagged phalloidin (Thermo Fisher Scientific) was used to stain actin stress fibers as per the manufacturer's recommendation.

## Immunohistochemistry

Tissues were fixed in 10% neutral buffered formalin for 24 hr and then transferred to PBS. Tissues were embedded in paraffin and 5 µm sections were processed for hematoxylin–eosin staining and immunohistochemistry. Antigen was retrieved by using citrate buffer at pH 6.0 at high heat and pressure for 30 min. Endogenous peroxidases were blocked with 3% hydrogen peroxide (10 min), followed by serum blocking (1 hr). Primary antibodies were incubated overnight at 4°C. Secondary antibodies (ImmPRESS Reagent Anti-Mouse IgG and Anti-Rabbit IgG from Vector Labs) were incubated at room temperature for 1 hr. Antigens were developed with ImmPACT DAB kit (Vector Labs) peroxidase substrate. Primary antibodies used were Mouse anti-p53 antibody (OP43, EMD Millipore, 1:100); Rabbit anti-CypD antibody (ab155979, abcam, 1:50).

## Mitochondrial fractionations

After cultured cells were trypsinized, counted, and washed with PBS, mitochondria were extracted from $10^7$ cells using a Mitochondria isolation kit (MACS Miltenyi Biotech, San Diego, California) containing anti-Tom22 mitochondria specific magnetic microbeads. After extraction, the mitochondria were lysed in buffer A (150 mM NaCl, 5 mM EDTA, 1% digitonin, and 50 mM Tris-HCl pH 7.5) for 1 hr. The amount of protein collected was quantified using a BCA kit, and samples were electrophoresed as described earlier. The purity of fractions was determined by immunoblotting with different cellular compartment markers.

## Immunoprecipitation

For immunoprecipitation mitochondrial fractions were lysed as described earlier. Lysed fractions were pre-cleared with agarose IgG beads for 30 min. FLAG tagged p53 was immunoprecipitated from 550 µg of the mitochondrial lysate with monoclonal Anti-FLAG M2 mouse antibody (Sigma-Aldrich, RRID: AB_262044) at 4°C overnight on a rotator. Samples were next washed 5 times with buffer B (150 mM NaCl, 5 mM EDTA, 0.5% digitonin, 1% triton X-100 and 50 mM Tris-HCl pH 7.5). The immunoprecipitated proteins were eluted and analyzed by immunoblotting for CypD binding.

## Mitochondrial permeability transition pore (MPTP) assay

To measure change in the mitochondrial permeability transition pore (MPTP) opening, p53 truncation-expressing cell lines were infected with p53 or CypD shRNA. Cells were collected post infection at 96 hr. Alterations in the functionality of MPTP were measured by assessing changes in calcein fluorescence by using MPTP kit (Biovision K239-100, Milpitas, California) as per the manufacturer's instructions. To measure MPTP opening, ectopically expressing p53 truncations cells were grown in 2% FBS for 36 hr and were treated with vehicle or 2 µM of CsA for 2 hr before collecting for analysis. To analyze the degree of pore opening quenching of calcein fluorescence by $CoCl_2$ is measured. As

shown below, higher the calcein fluorescence in presence of $CoCl_2$, more the decrease in pore permeability, and vice-versa.

## JC-1 assay

To measure the change in mitochondrial polarization, p53 truncations expressing cell lines were either infected with p53 or CypD shRNA. Cells were collected post-infection at 96 hr, and mitochondrial polarization was measured using the MitoProbe JC-1 assay kit manufacturer's instructions (Thermo Fisher Scientific-M34152).

## Drug sensitivity assay

For the drug sensitivity assay, an equal number (2000–8000 cells/well) of cells was plated in quadruplets in 24-well plates (BD falcon). After 24 hr, different concentrations of C-9 were added to the cells. At indicated time points, cells were washed with PBS, fixed with formaldehyde, stained with crystal violet and quantified as described earlier.

## Data analysis

All mutation data were obtained from the MSKCC cBioPortal (http://www.cbioportal.org). Statistical analyses were performed in R (http://cran.r-project.org/) (RRID:SCR_003005). Fisher's exact test was used to compare observed versus expected occurrence of various types of *TP53* mutation. Both nonsense mutations occurring in exon-6 and frameshift mutations wherein a premature stop codon was introduced in exon-6 were considered as exon-6 truncations.

All data and statistical analyses were performed using GraphPad Prism Software (RRID: SCR_002798).

## Acknowledgements

This study was supported by the NCI P01 CA129243-06 target for therapy for carcinomas in the lung, and Swim Across America. We would like to acknowledge Ms. Laura Maiorino and Dr. Robert Wysocki for technical assistance. We would like to acknowledge Drs. Chiara Gorrini, Tak Mak and Ute Moll for critical insights and discussions. We would also like to acknowledge Microscopy, Histology, Flow Cytometry and Laboratory Animal Resources core facilities at Cold Spring Harbor Laboratory.

## Additional information

### Funding

| Funder | Grant reference number | Author |
| --- | --- | --- |
| National Cancer Institute | NCI P01 CA129243-06 | Marc Ladanyi<br>Scott W Lowe<br>Raffaella Sordella |

The funders had no role in study design, data collection and interpretation, or the decision to submit the work for publication.

### Author contributions

NHS, DP, Conception and design, Acquisition of data, Analysis and interpretation of data, Drafting or revising the article; ERK, SS, MM, Acquisition of data, Analysis and interpretation of data, Drafting or revising the article; JB, Drafting or revising the article, Contributed unpublished essential data or reagents; PP, Acquisition of data, Analysis and interpretation of data; GM, Drafting or revising the article; MA, Acquisition of data, Contributed unpublished essential data or reagents; ML, Synthesized the small molecule C-9 and performed quality control assay to assure the purity of reagents, therefore his authorship in this study is very critical, Contributed unpublished essential data or reagents; SWL, Acquisition of data, Analysis and interpretation of data, Contributed unpublished essential data or reagents; RS, Analysis and interpretation of data, Drafting or revising the article, Contributed unpublished essential data or reagents

**Author ORCIDs**

Raffaella Sordella, http://orcid.org/0000-0001-9745-1227

**Ethics**

Animal experimentation: All animal experiments were performed in accordance with National Research Council's Guide for the Care and Use of Laboratory Animals. Protocols were approved by the Cold Spring Harbor Laboratory Animal Care and Use Committee (933922-1 Development of mouse models to study human lung cancer - integrated protocols).

## Additional files

### Supplementary files

• Supplementary file 1. List of tumor studies used for analysis of distribution of Missense, Exon-6 and other truncations in *TP53*.

• Supplementary file 2. Values of p53 mRNA expression with different mutations obtained from four different tumor types.

• Supplementary file 3. Number of tumor samples with indicated mutation types in primary and metastatic colorectal carcinoma.

• Supplementary file 4. Sequence of sense strand of shRNAs used in this study.

• Supplementary file 5. Complimentary oligonucleotides used for cloning the indicated sgRNAs.

### Major datasets

The following previously published datasets were used:

| Author(s) | Year | Dataset title | Dataset URL | Database, license, and accessibility information |
| --- | --- | --- | --- | --- |
| Cancer Genome Atlas Network | 2013 | Genomic and Epigenomic Landscapes of Adult De Novo Acute Myeloid Leukemia | http://www.cbioportal.org/index.do?Action=Submit&genetic_profile_ids=laml_tcga_pub_mutations&case_set_id=laml_tcga_pub_all&cancer_study_id=laml_tcga_pub&gene_list=TP53&tab_index=tab_visualize&#mutation_details&show_samples=false& | Publicly available at the cBioPortal (http://www.cbioportal.org). |

| | | | | |
|---|---|---|---|---|
| Ciriello G, Gatza ML, Beck AH, Wilkerson MD, Rhie SK, Pastore A, Zhang H, McLellan M, Yau C, Kandoth C, Bowlby R, Shen H, Hayat S, Fieldhouse R, Lester SC, Tse GM, Factor RE, Collins LC, Allison KH, Chen YY, Jensen K, Johnson NB, Oesterreich S, Mills GB, Cherniack AD, Robertson G, Benz C, Sander C, Laird PW, Hoadley KA, King TA; TCGA Research Network, Perou CM. | 2015 | Comprehensive Molecular Portraits of Invasive Lobular Breast Cancer | http://www.cbioportal.org/study?id=brca_tcga_pub2015#summary | Publicly available at the cBioPortal (http://www.cbioportal.org). |
| Cancer Genome Atlas Network | 2012 | Comprehensive molecular characterization of human colon and rectal cancer | http://www.cbioportal.org/study?id=coadread_tcga_pub#summary | Publicly available at the cBioPortal (http://www.cbioportal.org). |
| Dulak AM, Stojanov P, Peng S, Lawrence MS, Fox C, Stewart C, Bandla S, Imamura Y, Schumacher SE, Shefler E, McKenna A, Carter SL, Cibulskis K, Sivachenko A, Saksena G, Voet D, Ramos AH, Auclair D, Thompson K, Sougnez C, Onofrio RC, Guiducci C, Beroukhim R, Zhou Z, Lin L, Lin J, Reddy R, Chang A, Landrenau R, Pennathur A, Ogino S, Luketich JD, Golub TR, Gabriel SB, Lander ES, Beer DG, Godfrey TE, Getz G, Bass AJ. | 2013 | Exome and whole-genome sequencing of esophageal adenocarcinoma identifies recurrent driver events and mutational complexity. | http://www.cbioportal.org/study?id=esca_broad#summary | Publicly available at the cBioPortal (http://www.cbioportal.org). |
| Lin DC, Hao JJ, Nagata Y, Xu L, Shang L, Meng X, Sato Y, Okuno Y, Varela AM, Ding LW, Garg M, Liu LZ, Yang H, Yin 8, Shi ZZ, Jiang YY, Gu WY, Gong T, Zhang Y, Xu X, Kalid O, Shacham S, Ogawa S, Wang MR, Koeffler HP | 2014 | Genomic and molecular characterization of esophageal squamous cell carcinoma. | http://www.cbioportal.org/study?id=escc_ucla_2014#summary | Publicly available at the cBioPortal (http://www.cbioportal.org). |

| | | | | |
|---|---|---|---|---|
| Brennan CW, Verhaak RG, McKenna A, Campos B, Noushmehr H, Salama SR, Zheng S, Chakravarty D, Sanborn JZ, Berman SH, Beroukhim R, Bernard B, Wu CJ, Genovese G, Shmulevich I, Barnholtz-Sloan J, Zou L, Vegesna R, Shukla SA, Ciriello G, Yung WK, Zhang W, Sougnez C, Mikkelsen T, Aldape K, Bigner DD, Van Meir EG, Prados M, Sloan A, Black KL, Eschbacher J, Finocchiaro G, Friedman W, Andrews DW, Guha A, Iacocca M, O'Neill BP, Foltz G, Myers J, Weisenberger DJ, Penny R, Kucherlapati R, Perou CM, Hayes DN, Gibbs R, Marra M, Mills GB, Lander E, Spellman P, Wilson R, Sander C, Weinstein J, Meyerson M, Gabriel S, Laird PW, Haussler D, Getz G, Chin L; TCGA Research Network. | 2013 | The somatic genomic landscape of glioblastoma. | http://www.cbioportal. org/study?id=gbm_ tcga_pub2013#summary | Publicly available at the cBioPortal (http:// www.cbioportal. org). |
| Cancer Genome Atlas Network | 2015 | Comprehensive genomic characterization of head and neck squamous cell carcinomas. | http://www.cbioportal. org/study?id=hnsc_tcga_ pub#summary | Publicly available at the cBioPortal (http:// www.cbioportal. org). |
| Cancer Genome Atlas Network | 2014 | Comprehensive molecular profiling of lung adenocarcinoma | http://www.cbioportal. org/study?id=luad_tcga_ pub#summary | Publicly available at the cBioPortal (http:// www.cbioportal. org). |
| Cancer Genome Atlas Network | 2012 | Comprehensive genomic characterization of squamous cell lung cancers. | http://www.cbioportal. org/study?id=lusc_tcga_ pub#summary | Publicly available at the cBioPortal (http:// www.cbioportal. org). |
| Lohr JG, Stojanov P, Carter SL, Cruz-Gordillo P, Lawrence MS, Auclair D, Sougnez C, Knoechel B, Gould J, Saksena G, Cibulskis K, McKenna A, Chapman MA, Straussman R, Levy J, Perkins LM, Keats JJ, Schumacher SE, Rosenberg M; Multiple Myeloma Research Consortium, Getz G, Golub TR | 2014 | Widespread genetic heterogeneity in multiple myeloma: implications for targeted therapy. | http://www.cbioportal. org/study?id=mm_ broad#summary | Publicly available at the cBioPortal (http:// www.cbioportal. org). |

| | | | | |
|---|---|---|---|---|
| Beltran H, Prandi D, Mosquera JM, Benelli M, Puca L, Cyrta J, Marotz C, Giannopoulou E, Chakravarthi BV, Varambally S, Tomlins SA, Nanus DM, Tagawa ST, Van Allen EM, Elemento O1, 6, Sboner A, Garraway LA, Rubin MA, Demichelis F | 2016 | Divergent clonal evolution of castration-resistant neuroendocrine prostate cancer. | http://www.cbioportal. org/study?id=nepc_ wcm_2016#summary | Publicly available at the cBioPortal (http:// www.cbioportal. org). |
| Cancer Genome Atlas Network | 2011 | Integrated genomic analyses of ovarian carcinoma. | http://www.cbioportal. org/study?id=ov_tcga_ pub#summary | Publicly available at the cBioPortal (http:// www.cbioportal. org). |
| Cancer Genome Atlas Network | 2015 | The Molecular Taxonomy of Primary Prostate Cancer. | http://www.cbioportal. org/study?id=prad_ tcga_pub#summary | Publicly available at the cBioPortal (http:// www.cbioportal. org). |

| George J, Lim JS, Jang SJ, Cun Y, Ozretić L, Kong G, Leenders F, Lu X, Fernández-Cuesta L, Bosco G, Müller C, Dahmen I, Jahchan NS, Park KS, Yang D, Karnezis AN, Vaka D, Torres A, Wang MS, Korbel JO, Menon R, Chun SM, Kim D, Wilkerson M, Hayes N, Engelmann D, Pützer B, Bos M, Michels S, Vlasic I, Seidel D, Pinther B, Schaub P, Becker C, Altmüller J, Yokota J, Kohno T, Iwakawa R, Tsuta K, Noguchi M, Muley T, Hoffmann H, Schnabel PA, Petersen I, Chen Y, Soltermann A, Tischler V, Choi CM, Kim YH, Massion PP, Zou Y, Jovanovic D, Kontic M, Wright GM, Russell PA, Solomon B, Koch I, Lindner M, Muscarella LA, la Torre A, Field JK, Jakopovic M, Knezevic J, Castaños-Vélez E, Roz L, Pastorino U, Brustugun OT, Lund-Iversen M, Thunnissen E, Köhler J, Schuler M, Botling J, Sandelin M, Sanchez-Cespedes M, Salvesen HB, Achter V, Lang U, Bogus M, Schneider PM, Zander T, Ansén S, Hallek M, Wolf J, Vingron M, Yatabe Y, Travis WD, Nürnberg P, Reinhardt C, Perner S, Heukamp L, Büttner R, Haas SA, Brambilla E, Peifer M, Sage J, Thomas RK | 2015 | Comprehensive genomic profiles of small cell lung cancer. | http://www.cbioportal.org/study?id=sclc_uco-logne_2015#summary | Publicly available at the cBioPortal (http://www.cbioportal.org). |
|---|---|---|---|---|
| Cancer Genome Atlas Network | 2014 | Comprehensive molecular characterization of gastric adenocarcinoma. | http://www.cbioportal.org/study?id=stad_tcga_pub#summary | Publicly available at the cBioPortal (http://www.cbioportal.org). |

| Cancer Genome Atlas Research Network, Kandoth C, Schultz N, Cherniack AD, Akbani R, Liu Y, Shen H, Robertson AG, Pashtan I, Shen R, Benz CC, Yau C, Laird PW, Ding L, Zhang W, Mills GB, Kucherlapati R, Mardis ER, Levine DA | 2013 | Integrated genomic characterization of endometrial carcinoma. | http://www.cbioportal.org/study?id=ucec_tcga_pub#summary | Publicly available at the cBioPortal (http://www.cbioportal.org). |

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
