## [Decision Letter]

Thank you for submitting your article 'Tp53 exon-6 Truncating Mutations Produce Separation-of-Function Isoforms with Pro-tumorigenic Functions' for consideration by *eLife*. Your article has been reviewed by three peer reviewers, one of whom Joaquin Espinosa (Reviewer #1), is a member of our Board of Reviewing Editors and the evaluation has been overseen by Charles Sawyers as the Senior Editor. The following individuals involved in review of your submission have agreed to reveal their identity: Jean-Christophe Bourdon (Reviewer #2) and Antony Braithwaite (Reviewer #3).

The reviewers have discussed the reviews with one another and the Reviewing Editor has drafted this decision to help you prepare a revised submission.

Summary:

In this manuscript, Sordella and colleagues report the characterization of p53 exon 6 truncation mutants in human cancer cells. The manuscript contains a number of interesting observations, that could be summarized as follows:

1) Exon 6 nonsense (truncation) p53 mutants occur at a higher frequency relative to missense mutations, are stably expressed in many tumor cell lines (effectively bypassing nonsense-mediated decay) and, in the case of colorectal tumors, are more common in metastases than primary tumors.

2) Exon 6 nonsense p53 mutants harbor gain-of-function (GOF) activity, promoting cell viability, EMT-like properties in vitro and metastasis in a mouse model.

3) Exon 6 nonsense p53 mutants are mostly excluded from the nucleus, localizing instead to cytoplasm and mitochondria, where they bind to cyclophilin D (CypD) and promote mitochondrial membrane permeability.

4) Cancer cell lines expressing exon 6 nonsense mutants display dependency on both mutant p53 and CypD, suggesting an epistatic relationship between these proteins in promoting cell survival and EMT features.

Overall, the Reviewers found the paper interesting, fulfilling a gap in the p53 field, as p53 truncated mutants have not been explored in detail. The GOF mechanism reported was deemed interesting, because it is clearly different from that observed for p53 missense mutants, which act in the nucleus via transcriptional effects.

After thorough discussion, the Reviewers agreed to encourage submission of a revised manuscript addressing several major and minor points:

Essential points:

1) Reviewers agreed that a main conclusion in the manuscript is not fully supported by the data and that additional work is required to prove this point. The authors claim that p53 mutants lacking the DNA binding domain regulate mitochondrial membrane permeability (through MPTP activity) in a CypD-dependent manner. The data show that mutant p53 R213* binds to CypD but it is unclear whether this interaction occurs in the mitochondria or in the cytoplasm (if in the cytoplasm, it would imply that mutant p53 inhibits localization of CypD to the mitochondria which would explain its reduced mitochondrial localization in H1299 cells overexpressing R213* or R196*). Furthermore, it is unclear whether overexpression of CypD abolishes MPTP closure promoted by depletion of mutant p53 R213*. Therefore, the mechanistic and epistatic relationship between p53 truncated mutants and CypD must be explored in greater detail. Specific questions that must be answered experimentally are:

1.1) Is casein fluorescence (which indicates MPTP closure) dependent on CypD in the cell lines with truncated p53 (HoP62, DMS114, Calu-6 and SW684)?

1.2) Does knock-down of endogenous mutant p53 in cell lines expressing R213* or R196* change mitochondrial localization of CypD (HoP62, DMS114, Calu-6 and SW684 cell lines)?

1.3) Does knock-down of CypD change mitochondrial localization of endogenous mutant p53 in HoP62, DMS114, Calu-6 and SW684?

1.4) Does depletion of endogenous mutant p53 R213* or R196* in HoP62, DMS114, Calu-6 and SW684 change mitochondrial polarization and thus ATP production (this could be done by measuring fluorescence of JC-1 marker or rhodamine marker)?

2) All three reviewers commented on the underwhelming scholarship regarding the writing of the manuscript, the composition and explanation of the figures, and the interpretation of some results. The paper suffers from a lack of detail about the experimental procedures and discussion about the data, and overall poor writing with many typographical errors, poor sentence construction and obscure expressions. A lot of careful rewriting and editing needs to be done with the express goal of making the manuscript more readable and understandable. The following points must be carefully addressed in the revised manuscript:

2.1) Figure 1. Panel C –the point of this table needs to be clearer as all mutation types are 2-5 fold less than expected. Further, it looks like exon 6 truncations occur 10x less often than all mutations, not 5x more as stated (p4, para 2). Panel F – SLCA is not there and MM seems to follow the same trend as the other cancer types. Thus, the text in subsection 'TP53 mutations occur at higher than expected frequency.' (paragraph 5) is not accurate. Panel G – clearly, IF indel is significantly different in the metastatic group? Not NS as stated. Some comment should be made about this group.

2.2) Figure 2. Panel B – there are no size markers on the western blots; there are multiple protein species in cells transfected with mutants R306* and G325*. What are these? Is the R196* running as a doublet? Panel D – are these lentivirus-transduced cell lines or are they transiently transfected cells? If the latter, how were the transfected cells identified? The resolution of fluorescent images is poor, making it hard to see any morphological changes as described (subsection 'p53 exon-6 truncating mutants reprogram cells towards the acquisition of prometastatic Features', paragraph 2). Also, the DAPI stain cannot be seen properly in the R196*and R213* merges. Why is this? These images need to be improved or replaced. Why were A549 cells used rather than the H1299 p53 null cells? The null background avoids the confounder of p53 and its isoforms. Panel E – if the cells in D were transiently transfected, at which times were they assayed for EMT transcripts and were they always elevated? Panel F – it is hard to follow with the NS being on all box and whisper plots. Displaying this needs to be thought about more carefully. Also, again, why was the experiment not done in H1299 cells? Panel H – this does not measure metastasis; it measures colonization. The conclusions need to be altered. At what times were the mice harvested? Probably very soon after inoculation as B16 cells migrate rapidly to the lungs without any extra stimulus – more experimental details are required.

2.3) Figure 3. Panel A – Looking at the same data in the TCGA, whilst it is technically true that the transcript levels are similar for truncating compared to no mutation, the numbers of truncating mutations are very low (e.g. ~10 for sarcoma compared >100 with no mutation). This could be due to chance alone. This should be clarified or this panel should be deleted. Panel F – resolution is poor. Panel I – please define what Ren g 208 and p53g.140 is in the legend. Please provide evidence of knockdown.

2.4) Figure 4. This was generally hard to follow. More information in the text and/or the legend is needed. Panel F – please state clearly the p53 status of the Hop62 cell line. Panel G – why are the results with the 196* mutant not shown? Please explain what CsA does. Paragraph 4 in subsection 'p53 exon-6 mutations partially localize to the mitochondria where by interacting with Cyclophilin D control the mitochondria inner-pore permeability' about the W146* mutant is for the discussion, not the results, and it is poorly written. Please re-write for greater clarity.

2.5) Figure 5. Panel C – poor resolution. Panel F – please provide evidence that the DD-Cas9 actually worked. The right hand panel shows that KD of p53 increases tumor growth – this seems to be the opposite of what is stated in the text (subsection 'Cyclophilin D activity is required for phenotypes associated with p53 exon-6 truncating mutations', although the result is to be expected).

2.6) Discussion. The discussion needs to be written properly. The opening paragraph seems to be referring to the role of TP53 isoforms – 'multiplicity of alleles'. If so, the relevant references are wrong.

2.7) References. These are poorly done; authorship is seriously abbreviated and there are several errors.

---

## [Author Response]

[…]

*Essential points:*

*1) Reviewers agreed that a main conclusion in the manuscript is not fully supported by the data and that additional work is required to prove this point. The authors claim that p53 mutants lacking the DNA binding domain regulate mitochondrial membrane permeability (through MPTP activity) in a CypD-dependent manner. The data show that mutant p53 R213* binds to CypD but it is unclear whether this interaction occurs in the mitochondria or in the cytoplasm (if in the cytoplasm, it would imply that mutant p53 inhibits localization of CypD to the mitochondria which would explain its reduced mitochondrial localization in H1299 cells overexpressing R213* or R196*). Furthermore, it is unclear whether overexpression of CypD abolishes MPTP closure promoted by depletion of mutant p53R213*. Therefore, the mechanistic and epistatic relationship between p53 truncated mutants and CypD must be explored in greater detail. Specific questions that must be answered experimentally are:*

*1.1) Is casein fluorescence (which indicates MPTP closure) dependent on CypD in the cell lines with truncated p53 (HoP62, DMS114, Calu-6 and SW684)?*

As shown in Figure 4—figure supplement 4, knockdown of CypD by shRNA in Hop62, SW684, DMS114 and Calu-6 results in an increase in the percentage of calcein positive cells which, in turn, suggests the closure of the MPTP upon decreased expression of CypD. However, this difference was not observed in cells expressing p53 wild-type (A549).

*1.2) Does knock-down of endogenous mutant p53 in cell lines expressing R213* or R196* change mitochondrial localization of CypD (HoP62, DMS114, Calu-6 and SW684 cell lines)?*

Western blot analysis of total and mitochondria cell extracts indicated that knockdown of endogenous p53 in Hop62, DMS114, SW684 and Calu-6 cell lines did not affect the overall levels of CypD nor its mitochondrial localization. See Figure 4—figure supplement 6.

*1.3) Does knock-down of CypD change mitochondrial localization of endogenous mutant p53 in HoP62, DMS114, Calu-6 and SW684?*

Also in this case, we did not observe any difference in the levels of expression nor in the mitochondrial localization of p53 upon knockdown of CypD with two independent shRNAs. See Figure 4—figure supplement 7.

*1.4) Does depletion of endogenous mutant p53 R213* or R196* in HoP62, DMS114, Calu-6 and SW684 change mitochondrial polarization and thus ATP production (this could be done by measuring fluorescence of JC-1 marker or rhodamine marker)?*

As suggested by the reviewers, we utilized the JC-1 assay to evaluate changes in mitochondrial polarization upon p53 and CypD knockdown in A549, Hop62, DMS114, Calu-6 and SW684 cells. We found that reducing the expression of p53 or CypD in cell lines expressing p53-psi or p53 exon-6 truncations resulted in an increase in J aggregates. See Figure 4 and Figure 4—figure supplement 4.

Although changes in the mitochondrial polarization can impact ATP production, several other metabolic pathways can affect net ATP production in cells. By only measuring ATP levels and without conducting a detailed analysis of metabolic flow in cells that supersedes the intended purpose of this work, it will be difficult to determine whether p53-psi and p53-psi like mutations affect the ATP production via deregulation of mitochondrial membrane potential.

*2) All three reviewers commented on the underwhelming scholarship regarding the writing of the manuscript, the composition and explanation of the figures, and the interpretation of some results. The paper suffers from a lack of detail about the experimental procedures and discussion about the data, and overall poor writing with many typographical errors, poor sentence construction and obscure expressions. A lot of careful rewriting and editing needs to be done with the express goal of making the manuscript more readable and understandable. The following points must be carefully addressed in the revised manuscript:*

We apologize for this. We are surprised as the manuscript was edited by an external editing company (NPG-editing). Based on these comments, we have carefully reexamined the original manuscript, included additional details about the experimental procedures and expanded upon our discussion of the results.

*2.1) Figure 1. Panel C – the point of this table needs to be clearer as all mutation types are 2-5 fold less than expected. Further, it looks like exon 6 truncations occur 10x less often than all mutations, not 5x more as stated (p4, para 2). Panel F – SLCA is not there and MM seems to follow the same trend as the other cancer types. Thus, the text in subsection 'TP53 mutations occur at higher than expected frequency.' (paragraph 5) is not accurate. Panel G – clearly, IF indel is significantly different in the metastatic group? Not NS as stated. Some comment should be made about this group.*

– Panel C. The table compares the theoretical (e.g., expected frequency) of mutations with the frequencies of mutations we observed (e.g., observed frequency) in the TCGA and MSKCC datasets. In the case of missense mutations, we estimated an expected frequency of 0.34. In both the TCGA and the MSKCC cohorts, the observed frequency was 0.64 and 0.52, respectively. Therefore, we calculated the frequency of missense mutations observed in tumors was 1.5 and 1.9 fold higher than the expected frequencies in the TCGA and in the MSKCC datasets respectively. As for nonsense mutations, the expected frequency was 0.054 while the observed frequency was 0.084 in the TCGA data set and 0.076 in the MSKCC data set. This was not the case for exon-6 nonsense mutations. In fact, we found that while the expected frequency was 0.0098, the observed frequency was 0.05 in the TCGA cohort and 0.04 in the MSKCC cohort.

Altogether these data indicate that the exon-6 nonsense mutations, differently from missense or other non-sense mutations are 4-5 fold more frequent than expected (Figure 1).

– Panel F. Small cell lung carcinoma was indicated as LUSCC in the Figure 1 and [Supplementary-material SD1-data]. As indicated in [Supplementary-material SD1-data], 'MM' stands for multiple myeloma. In this tumor type, the frequency of mutation of *TP53* and exon-6 mutations are rare. We are not sure about the comment of the reviewer as multiple myeloma was not refer to in the text.

– Panel G. While the ratio between IF-indel is higher in the metastatic group, the low number of samples with primary tumors (n= 3 primary and n= 10 metastatic tumor site) rendered this finding statistically insignificant, as indicated.

*2.2) Figure 2. Panel B – there are no size markers on the western blots; there are multiple protein species in cells transfected with mutants R306* and G325*. What are these? Is the R196* running as a doublet? Panel D – are these lentivirus-transduced cell lines or are they transiently transfected cells? If the latter, how were the transfected cells identified? The resolution of fluorescent images is poor, making it hard to see any morphological changes as described (subsection 'p53 exon-6 truncating mutants reprogram cells towards the acquisition of prometastatic Features', paragraph 2). Also, the DAPI stain cannot be seen properly in the R196*and R213* merges. Why is this? These images need to be improved or replaced. Why were A549 cells used rather than the H1299 p53 null cells? The null background avoids the confounder of p53 and its isoforms. Panel E – if the cells in D were transiently transfected, at which times were they assayed for EMT transcripts and were they always elevated? Panel F – it is hard to follow with the NS being on all box and whisper plots. Displaying this needs to be thought about more carefully. Also, again, why was the experiment not done in H1299 cells? Panel H – this does not measure metastasis; it measures colonization. The conclusions need to be altered. At what times were the mice harvested? probably very soon after inoculation as B16 cells migrate rapidly to the lungs without any extra stimulus – more experimental details are required.*

*–* Panel B.Molecular weight markers have been added onto all of the western blots. We are not clear on the nature of the additional bands observed, and believe they are most likely the result of proteolysis. Interestingly, western blot analysis in B16 cells did not show these extra-bands. Further studies will be required to better understand the nature and a possible functional role of these additional p53 species.

*–* Panel D. The cell lines were generated using a lentiviral ectopic expression system described in the materials and methods section. We have now replaced the original image and provided a better resolution with particular attention to DAPI levels.

The reason why we used A549 cells to illustrate differences in the morphology of cells upon ectopic expression of p53 mutants was mainly based on the fact that A549 cells form a nice columnar epithelial monolayer when they reach confluence. This facilitates the analysis of the distribution of cortical actin, stress fibers and E-cadherin. Nevertheless, as shown in in Figure 2—figure supplement 3, also in the H1299 cells we observed that ectopic expression of R196* and R213* induced an increase in the expression of mesenchymal markers (p-value *<0.05, unpaired t-test).

*–* Panel E. The cells were generated using a constitutive lentiviral system. We assayed the cells for expression of mesenchymal markers 96 hours after seeding. We did not observe much difference across experiments.

*–* Panel F. As per the reviewers’ suggestions, we changed the whisker plots and we have extended our analysis also to a p53 null cell line (B16-F1). See Figure 2, Figure 2—figure supplement 1.

*–* Panel H. Although in the literature this model is referred to as metastatic assay (Overwijk and Restifo, 2001) we agree with the reviewers that this assay more accurately reflects colonization potential and not metastatic spread. Therefore, we integrated the suggested changes in the text. Experimental details have been added in the Materials and Methods section.

*2.3) Figure 3. Panel A – Looking at the same data in the TCGA, whilst it is technically true that the transcript levels are similar for truncating compared to no mutation, the numbers of truncating mutations are very low (e.g. ~10 for sarcoma compared >100 with no mutation). This could be due to chance alone. This should be clarified or this panel should be deleted. Panel F – resolution is poor. Panel I – please define what Ren g 208 and p53g.140 is in the legend. Please provide evidence of knockdown.*

*–* Panel A. To confirm that these observations were not due to chance alone and to strengthen our point, we have added a summary table and performed a Student’s t-test with FDR of 5%. As shown in the table below, the p-values confirm the statistical significance of our observations.

Tumor typeNo mutationTruncating mutationp-valuePancreas (TCGA)45300.0001684Sarcoma (TCGA)204633.18365E-06Lung Squamous (TCGA)36420.001551394Head & Neck (TCGA)77711.2E-14

*–* Panel F. We have substituted the panel for a picture with improved resolution as shown in Figure 3.

– Panel I. We have added additional information describing Ren g.208 and p53 g.140 in the figure legend, and we have provided the sequence of the guide RNAs used in this study in [Supplementary-material SD5-data]. We validated the inactivation of p53 by immunoblot (Figure 3—figure supplement 7) and by immunohistochemistry (Figure 3—figure supplement 7).

*2.4) Figure 4. This was generally hard to follow. More information in the text and/or the legend is needed. Panel F – please state clearly the p53 status of the Hop62 cell line. Panel G – why are the results with the 196* mutant not shown? Please explain what CsA does. Paragraph 4 in subsection 'p53 exon-6 mutations partially localize to the mitochondria where by interacting with Cyclophilin D control the mitochondria inner-pore permeability' about the W146* mutant is for the discussion, not the results, and it is poorly written. Please re-write for greater clarity.*

We have substantially amended this section, adding the p53 status of Hop62 cells, results for R196* and information regarding CsA. We have also moved the second paragraph to the Discussion section as suggested.

*2.5) Figure 5. Panel C – poor resolution. Panel F – please provide evidence that the DD-Cas9 actually worked. The right hand panel shows that KD of p53 increases tumor growth – this seems to be the opposite of what is stated in the text (subsection 'Cyclophilin D activity is required for phenotypes associated with p53 exon-6 truncating mutations', although the result is to be expected).*

*–* Panel C. We have improved the resolution of panel C and Figure 5.

*–* Panel F. We provided evidence of CypD inactivation in Figure 5—figure supplement 2 (immunoblot) and Figure 5—figure supplement 2 (immunohistochemistry).

We are not sure about the comment of the reviewer regarding the effect of knockdown on tumor growth. As shown in the chart in Figure 5, CypD knockdown in Calu-6 (R196*) reduced tumor volume as expected. The panel on the right shows changes in tumor volume in p53-WT (A549) cells. Interestingly, in this case we observed increased tumor growth. The latter does not modify the conclusion of our study indicating that cells harboring p53 exon-6 truncation are 'addicted' to CypD, but could have clinical/therapeutic implications as it argues the need of stratifying patients for p53 mutations when considering treatments based on CypD inhibition.

*2.6) Discussion. The discussion needs to be written properly. The opening paragraph seems to be referring to the role of TP53 isoforms – 'multiplicity of alleles'. If so, the relevant references are wrong.*

As previously indicated, we have revised the discussion and updated references. In the discussion we refer 'multiplicity of alleles' to different p53 mutant species conferring pro-tumorigenic properties.